# Systematic substrate identification indicates a central role for the metalloprotease ADAM10 in axon targeting and synapse function

Peer-Hendrik Kuhn[1,2,3]*, Alessio Vittorio Colombo[1,4], Benjamin Schusser[5], Daniela Dreymueller[6], Sebastian Wetzel[7], Ute Schepers[8], Julia Herber[1,4], Andreas Ludwig[6], Elisabeth Kremmer[9], Dirk Montag[10], Ulrike Müller[11], Michaela Schweizer[12], Paul Saftig[7], Stefan Bräse[8], Stefan F Lichtenthaler[1,3,4,13]*

[1]Neuroproteomics, Klinikum rechts der Isar, Technische Universität München, Munich, Germany; [2]Institut für Pathologie und Pathologische Anatomie, Technische Universität München, Munich, Germany; [3]Institute for Advanced Study, Technische Universität München, Munich, Germany; [4]Deutsches Zentrum für Neurodegenerative Erkrankungen, Munich, Germany; [5]Department of Animal Science, Institute for Animal Physiology, Ludwig-Maximilians-Universität München, Munich, Germany; [6]Institute of Pharmacology and Toxicology, Uniklinik RWTH Aachen, Aachen, Germany; [7]Institute of Biochemistry, Christian-Albrechts Universität zu Kiel, Kiel, Germany; [8]Karlsruhe Institute of Technology, Karlsruhe, Germany; [9]German Research Center for Environmental Health, Institute of Molecular Tumor immunology, Helmholtz Zentrum München, Munich, Germany; [10]Neurogenetics, Leibniz Institute for Neurobiology, Magdeburg, Germany; [11]Department of Functional Genomics, Institute for Pharmacy and Molecular Biotechnology, Heidelberg University, Heidelberg, Germany; [12]Service-Gruppe für Elektronenmikroskopie, Zentrum für Molekulare Neurobiologie, Hamburg, Germany; [13]Munich Cluster for Systems Neurology, Munich, Germany

*For correspondence:
peerhendrik@gmx.net (PHK);
stefan.lichtenthaler@dzne.de (SFL)

**Competing interests:** The authors declare that no competing interests exist.

**Abstract** Metzincin metalloproteases have major roles in intercellular communication by modulating the function of membrane proteins. One of the proteases is the a-disintegrin-and-metalloprotease 10 (ADAM10) which acts as alpha-secretase of the Alzheimer's disease amyloid precursor protein. ADAM10 is also required for neuronal network functions in murine brain, but neuronal ADAM10 substrates are only partly known. With a proteomic analysis of Adam10-deficient neurons we identified 91, mostly novel ADAM10 substrate candidates, making ADAM10 a major protease for membrane proteins in the nervous system. Several novel substrates, including the neuronal cell adhesion protein NrCAM, are involved in brain development. Indeed, we detected mistargeted axons in the olfactory bulb of conditional ADAM10-/- mice, which correlate with reduced cleavage of NrCAM, NCAM and other ADAM10 substrates. In summary, the novel ADAM10 substrates provide a molecular basis for neuronal network dysfunctions in conditional ADAM10-/- mice and demonstrate a fundamental function of ADAM10 in the brain.

**eLife digest** Several neurodegenerative disorders, including Alzheimer's disease, arise when protein-cutting enzymes process proteins in the wrong way. The resulting protein fragments can accumulate in nerve cells and cause them to die, leading to symptoms such as memory loss. In the case of Alzheimer's disease the toxic protein fragment – called amyloid beta – can be produced when one enzyme cuts the amyloid precursor protein. However, the amyloid beta fragment is not made when a different enzyme called ADAM10 cuts the amyloid precursor protein first.

There has been a lot of interest in finding drugs that activate ADAM10 to treat Alzheimer's disease. However, ADAM10 also cuts other proteins on the surface of cells and it is important to know about these proteins if ADAM10 is going to be successfully targeted by a drug.

To tackle this issue, Kuhn et al. have now searched for new proteins (or 'substrates') that are cut by ADAM10 in mouse nerve cells. The experiments identified proteins that were cut in normal nerve cells, but remained unprocessed in cells where the gene for ADAM10 had been deleted. This search uncovered almost 100 new substrates of ADAM10 that were then validated using biochemical techniques.

Among these substrates were many proteins that are normally anchored into the membranes of nerve cells and involved in guiding and positioning these cells in the brain so that they can connect and communicate with each other. Kuhn et al. then deleted the gene for ADAM10 only in the frontmost part of the mouse brain. This led to the nerve cells forming abnormal networks in the regions of the brain that process smells and emotions. Overall the experiments proved that ADAM10 is important not only for the prevention of Alzheimer's disease, but also for the normal development of the brain. Future studies could now explore how stimulating ADAM10 affects the levels of its substrates. Also, a better understanding of the substrates of ADAM10 may be useful both to predict side effects of drugs that activate ADAM10 and to monitor patients who are responding well to these drugs.

## Introduction

Proteolysis of cell surface membrane proteins is a basic cellular mechanism that controls intercellular communication and the interaction of cells with their extracellular environment. Proteolysis typically occurs close to the luminal or extracellular side of the membrane and results in release of the membrane protein's ectodomain. This process, referred to as ectodomain shedding, is a mechanism that shapes the cell surface by controlling the ectodomain length of cell surface membrane proteins thus modulating their function (*Weber and Saftig, 2012*).

Membrane-bound metalloproteases of the metzincin family have a key role in catalyzing ectodomain shedding of membrane proteins, in particular members of the 'a disintegrin and metalloprotease' (ADAM) subfamily (*Weber and Saftig, 2012*). One of them is ADAM10, which is a ubiquitously expressed type I membrane protein whose active site is located within its ectodomain, well positioned to shed the ectodomains of its substrates (*Weber and Saftig, 2012*). A key substrate for ADAM10 is the Notch receptor, which requires ADAM10-mediated shedding for its signaling during differentiation and development (*Hartmann et al., 2002*; *Qi et al., 1999*). Consequently, constitutive ADAM10-deficient mice die at embryonic day 9.5 most likely due to a loss of Notch signaling (*Hartmann et al., 2002*). Another major substrate of ADAM10 is the amyloid precursor protein (APP) for which ADAM10 acts as the constitutive alpha-secretase (*Postina et al., 2004*; *Lammich et al., 1999*; *Kuhn et al., 2010*; *Jorissen et al., 2010*) and thus possesses the ability to prevent the generation of the pathogenic Aβ peptide in Alzheimer's disease (AD) (*Lammich et al., 1999*; *Kuhn et al., 2010*). This makes ADAM10 a major drug target for AD (*Postina et al., 2004*), and an activator of ADAM10 expression has been tested in a clinical trial for AD (*Endres et al., 2014*). Whether such a therapeutic approach is safe, remains to be seen, in particular because relatively little is known about ADAM10 substrates in brain.

Besides Notch and APP, additional ADAM10 substrates, such as E-cadherin and CX3CL1, have been identified in different organs and tissues (*Hundhausen et al., 2003*). Given the embryonically lethal phenotype of constitutive ADAM10-deficient mice, little is known about ADAM10 substrates

in the central nervous system (*Hartmann et al., 2002*). A few ADAM10 substrates have been identified in the brain. Some of them have neuronal and synaptic functions, such as APP (*Ring et al., 2007*; *Weyer et al., 2011*; *2014*), Neuroligin-1 (*Blundell et al., 2010*; *Kim et al., 2008*) or N-Cadherin, which is in line with the phenotypes reported for conditional ADAM10-/- mice lacking ADAM10 expression in most neurons (*Jorissen et al., 2010*; *Gibb et al., 2010*; *Prox et al., 2013*). While these mice circumvent ADAM10 dependent embryonic lethality, they show epileptic seizures, learning deficits and an altered morphology of postsynaptic structures in the brain and die postnatally. This demonstrates that ADAM10 is essential for synaptic and neuronal network functions in the mouse brain (*Jorissen et al., 2010*; *Prox et al., 2013*).

Yet, the known ADAM10 substrates only partly explain these phenotypes and more ADAM10 substrates are expected to exist in brain. Their identification will allow a better mechanistic understanding of ADAM10 function in brain. Moreover, new ADAM10 substrates may be useful as biomarkers to evaluate how a patient responds to an ADAM10-modulating drug, for example in clinical trials for AD.

To systematically identify neuronal ADAM10 substrates, we used the quantitative proteomic 'secretome protein identification with click sugars' (SPECS) method, which has already been successfully applied to the identification of substrates for the membrane proteases BACE1 and SPPL3 (*Kuhn et al., 2012*; *2015*). SPECS allows specific enrichment of cell-derived glycoproteins which otherwise would escape detection by mass spectrometry due to their low abundance. Using high-resolution mass spectrometry and label-free quantification we systematically analyzed both the levels of membrane proteins in the neuronal membrane as well as their shed ectodomains in the secretome of primary, murine neurons, either expressing or lacking ADAM10. We identified 91, mostly novel ADAM10 substrate candidates.

Selected substrates, including NRCAM, LDLR, MT4MMP and CDH6 were validated by quantitative immunoblots in neurons and in mouse brain and point to a central function of ADAM10 in synapse formation and axon targeting. In line with this we detected an axon targeting defect in the olfactory bulb of conditional ADAM10-/- mice, similar to what has been observed in mice deficient in the novel ADAM10 substrate NRCAM.

## Results

### Identification of neuronal ADAM10 substrate candidates

To identify novel ADAM10 substrates we quantitatively compared the secretome of neurons with and without ADAM10 activity following the rationale that a lack in ADAM10 activity would reduce or almost abolish ectodomain shedding of a defined fraction of single span membrane proteins with other membrane proteins and soluble proteins being unaffected. To this aim, we isolated and cultured primary neurons from E15/E16 brains of a conditional Adam10 knockout mouse model (Ad10 fl/fl) allowing an Adam10 specific knockout upon iCre expression (*Gibb et al., 2010*). After two days in vitro (DIV) neurons were either transduced with an empty control lentivirus or a lentivirus that coded for iCre (*Figure 1A*). Two days later, neurons were incubated for another two days with the chemically modified sugar ManNAz, which is metabolized and incorporated as sialic acid into the glycan moieties of newly synthesized, cellular glycoproteins. Staining for beta III tubulin suggests that knockout of ADAM10 did not alter neuronal differentiation of primary neurons in vitro (*Figure 1A*). Media of both experimental conditions (conditioned for 48 hr) were processed with the SPECS method to selectively enrich the endogenous, cellular glycoproteins in the medium. Considering at least two unique peptides per protein and detection of the protein (protein group) in at least 4 out of 5 biological replicates the mass spectrometric analysis identified 313 proteins annotated as glycoproteins. For every glycoprotein we mapped all identified peptides to its extracellular, transmembrane and cytoplasmic domains, using the QARIP webserver (*Ivankov et al., 2013*). The majority of peptides matched to the extracellular domain, but not to the transmembrane or cytoplasmic domains (data not shown), which demonstrates that we had detected the proteolytically released ectodomain and not the full-length membrane proteins in the neuronal secretome.

We classified these glycoproteins according to their topology (*Figure 1B*). These comprised 113 secreted proteins and 200 membrane proteins of which 108 had type I orientation (*Figure 1B*). Proteins with other membrane orientations or with a GPI-anchor were also detected (*Figure 1B*) which

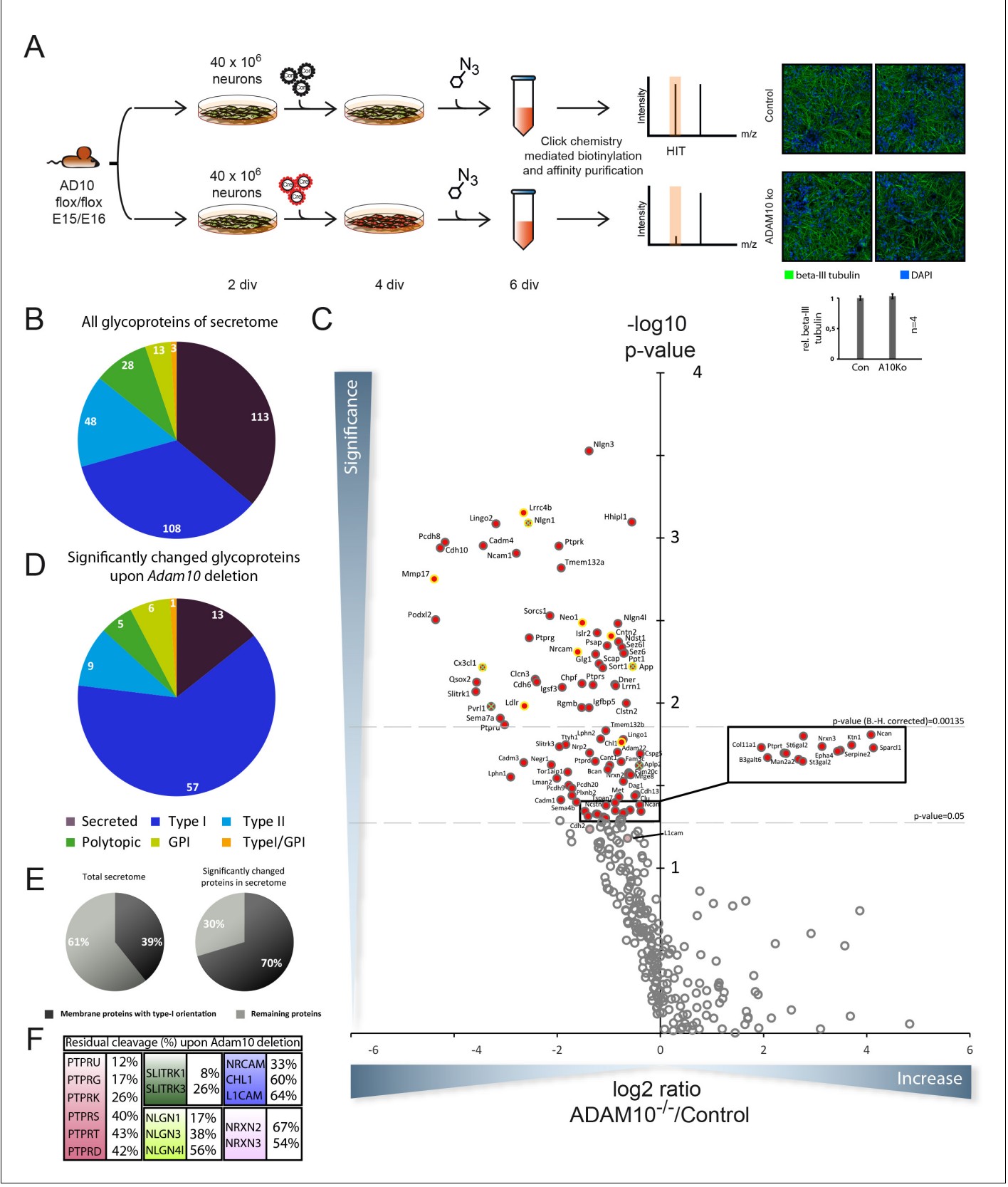

**Figure 1.** Identification of candidate ADAM10 substrates in conditional ADAM10-/- neurons. (**A**) Workflow for the identification of ADAM10 substrates in the secretome of primary cortical neurons comprising plating of neurons, lentiviral transduction, metabolic glycan labeling, purification and protein

*Figure 1 continued on next page*

*Figure 1 continued*

identification and quantification via mass spectrometry. Staining of primary cortical neurons for the neuron specific marker beta III tubulin and the nuclear stain DAPI followed by quantification of beta III tubulin staining supports that neuronal differentiation was unaffected by deletion of ADAM10. (B) Topology of all glycoproteins in the secretome. (C) Volcano plot of the quantitative comparison between the secretomes of wild type (wt) and Adam10 knockout neurons of all in at least 4 out of 5 experiments identified glycoproteins in the secretome of *Adam10-/-* and wt neurons. The p-value is depicted as negative decadic logarithm while the fold-change (Adam10-/-/wt) is depicted as log2 value. Significantly changed protein: p≤0,05 (-log10≥1,3). Substrates that are significantly reduced are marked red. Proteins which were validated with immunoblot have a yellow ring. Proteins which have been formerly described in literature as ADAM10 substrates are marked with a green cross. Hits whose p-values survived correction for multiple hypothesis testing yielded a p-value of at least 0.00135 as new significance niveau. (D) Topology of all significantly changed proteins in the secretome between Adam10-/- and wt neurons. (E) Percentage of all membrane proteins with type-I topology in the total secretome and among the significantly changed proteins upon Adam10 knockout. (F) Remaining shedding after Adam10 deletion in percent for members of selected protein families like the Neurexin (NRXN), Neuroligin (NLGN), receptor tyrosine phosphatase (PTPR), Slit and receptor tyrosine kinase domain (SLITRK) and the L1 family.

is similar to the data in our seminal SPECS study that identified BACE1 substrates in the neuronal secretome (*Kuhn et al., 2012*).

Label-free-quantification of our mass spectrometric analysis revealed that the levels of 91 glycoproteins were significantly reduced in the secretome upon Adam10 deletion considering a p-value cut off of 0.05 based on 5 biological replicates (*Figure 1C*, *Supplementary file 1* – A10 knockout quantified secretome data set). When applying false-discovery rate (FDR)-based multiple hypothesis testing according the method of Benjamini and Hochberg considering an FDR = 0.1 and all identified glycoproteins as hypotheses, 46 membrane proteins remain significantly reduced (*Figure 1C* and *Table 1*).

Surprisingly, not a single glycoprotein was significantly increased in the secretome upon Adam10 deletion. Topological classification of the 91 glycoproteins (p<0.05) with reduced levels in the Adam10-/- neuronal secretome revealed a strong enrichment of type-I membrane proteins (*Figure 1B*). Among the 91 significantly changed glycoproteins we identified 78 membrane proteins comprising 57 type-I, 6 GPI-anchored, 1 type-I/GPI-anchored, 5 polytopic and 9 type-II membrane proteins which we considered as potential ADAM10 substrate candidates due to their membrane localization (*Supplementary file 1* – A10 knockout quantified secretome dataset).

Among the significantly reduced proteins upon Adam10 deletion, type-I membrane proteins were enriched to 70% compared to the total secretome that contains only 39% type-I membrane protein (*Figure 1E*). This enrichment of membrane proteins with a type-I orientation was even more pronounced when we assigned the topology to the 42 most significantly reduced glycoproteins upon Adam10 deletion in the secretome of which 90% possess a type-I orientation (*Table 1*), which is in line with the observation that most previously identified ADAM10 substrates also have a type I orientation (*Weber and Saftig, 2012*).

Among all identified ADAM10 substrate candidates we found previously described ADAM10 substrates like Neuroligin-1 (NLGN1), fractalkine (CX3CL1) or the amyloid precursor protein (APP) (*Figure 1C*, yellow ring with green cross) which validated our experimental approach (*Lammich et al., 1999*; *Kuhn et al., 2010*; *Hundhausen et al., 2003*; *Suzuki et al., 2012*; *Reiss et al., 2005*; *Colombo et al., 2013*). The formerly known ADAM10 substrate N-Cadherin (CDH2) was borderline significant due to one outlier. Our SPECS-based substrate identification was characterized by high sensitivity as even small reductions in total shedding were detected in ADAM10-/- cells. For example, in neurons APP is mostly shed by the aspartyl protease BACE1 and only to a low extent by ADAM10 (*Colombo et al., 2013*). Yet a 20% reduction in APP shedding upon Adam10 deletion was clearly detected in our SPECS analysis. Interestingly, some substrates, such as Neuroligin-1 are 'exclusive' ADAM10 substrates, as their shedding was nearly completely abolished upon ADAM10 deletion. However, for other substrates including APP and CHL1, shedding was only partly reduced in ADAM10-/- neurons, indicating that they are not only substrates for ADAM10, but also for other proteases.

The list of identified ADAM10 substrate candidates comprises entire receptor families involved in synapse function and formation and axon targeting (*Figure 1F*). This became apparent for example for the receptor tyrosine phosphatase (Ptpr), the Neuroligin (NLGN), the Neurexin (NRXN) and the SLIT and NTRK (Slitrk) families, which are involved in synapse function, and the L1 adhesion molecule family, which is involved in axon targeting. In case of the receptor tyrosine phosphatase family

**Table 1.** Proteins that are significantly reduced in the secretome upon Cre recombinase induced deletion of ADAM10. The table contains the 42 most reduced proteins upon ADAM10 deletion. Indicated are the names of the proteins, the gene name, number of unique peptides, topology, the mean of the ratio between neurons devoid of ADAM10 and neurons expressing endogenous levels of ADAM10 of 5 biological replicates and the p-value calculated with a two-sided, heteroscedastic t-test based on the intensity ratios for the control and the ADAM10 deletion condition. Protein names of previously published ADAM10 substrates are highlighted in italic bold. (MEAN) Mean value of all 5 biological replicates, (SEM) Standard error of the mean. (Peptides) Number of peptides identified for every protein. Gene symbols of proteins validated with immunoblot are marked bold.

| Protein Name | Gene Symbol | Peptides | Topology | MEAN (A10KO/CON) | TTEST |
|---|---|---|---|---|---|
| Matrix metalloproteinase-17 | **Mmp17** | 3 | GPI | 0,05 | 1,77E-03 |
| Podocalyxin-like protein 2 | Podxl2 | 3 | Type-I | 0,05 | 3,13E-03 |
| Cadherin-10 | Cdh10 | 4 | Type-I | 0,05 | 1,15E-03 |
| Protocadherin-8 | Pcdh8 | 16 | Type-I | 0,06 | 1,06E-03 |
| SLIT and NTRK-like protein 1 | Slitrk1 | 4 | Type-I | 0,08 | 8,52E-03 |
| Sulfhydryl oxidase 2 | Qsox2 | 3 | Type-I | 0,08 | 7,47E-03 |
| *Fractalkine* | **Cx3cl1** | 2 | Type-I | 0,09 | 6,08E-03 |
| Cell adhesion molecule 4 | Cadm4 | 5 | Type-I | 0,09 | 1,11E-03 |
| *Poliovirus receptor-related protein 1* | Pvrl1 | 3 | Type-I | 0,10 | 1,05E-02 |
| Leucine-rich repeat and immunoglobulin-like domain-containing nogo | Lingo2 | 2 | Type-I | 0,11 | 8,21E-04 |
| Semaphorin-7A | Sema7a | 8 | GPI | 0,12 | 1,24E-02 |
| Receptor-type tyrosine-protein phosphatase U | Ptpru | 6 | Type-I | 0,12 | 1,35E-02 |
| Latrophilin-1 | Lphn1 | 7 | Type-I | 0,13 | 2,81E-02 |
| *Neural cell adhesion molecule 1* | Ncam1 | 16 | Type-I/GPI | 0,14 | 1,24E-03 |
| Leucine-rich repeat-containing protein 4B | **Lrrc4b** | 16 | Type-I | 0,16 | 7,03E-04 |
| Cell adhesion molecule 3 | Cadm3 | 5 | Type-I | 0,16 | 2,30E-02 |
| Low-density lipoprotein receptor | **Ldlr** | 13 | Type-I | 0,16 | 1,04E-02 |
| *Neuroligin-1* | **Nlgn1** | 5 | Type-I | 0,17 | 8,12E-04 |
| Receptor-type tyrosine-protein phosphatase gamma | Ptprg | 4 | Type-I | 0,17 | 4,03E-03 |
| H(+)/Cl(-) exchange transporter 3 | Clcn3 | 2 | Polytopic | 0,19 | 7,17E-03 |
| Cadherin-6 | **Cdh6** | 5 | Type-I | 0,19 | 7,46E-03 |
| VPS10 domain-containing receptor SorCS1 | Sorcs1 | 3 | Type-I | 0,23 | 2,96E-03 |
| Neuronal growth regulator 1 | Negr1 | 3 | GPI | 0,23 | 2,37E-02 |
| Vesicular integral-membrane protein VIP36 | Lman2 | 3 | Type-I | 0,25 | 2,86E-02 |
| Receptor-type tyrosine-protein phosphatase kappa | Ptprk | 12 | Type-I | 0,26 | 1,12E-03 |
| SLIT and NTRK-like protein 3 | Slitrk3 | 3 | Type-I | 0,26 | 1,84E-02 |
| Cell adhesion molecule 1 | Cadm1 | 6 | Type-I | 0,26 | 3,85E-02 |
| Transmembrane protein 132A | Tmem132a | 2 | Type-I | 0,26 | 1,52E-03 |
| Immunoglobulin superfamily member 3 | Igsf3 | 25 | Type-I | 0,27 | 8,04E-03 |
| Protein tweety homolog 1 | Ttyh1 | 5 | Type-I | 0,28 | 1,79E-02 |
| Torsin-1A-interacting protein 1 | Tor1aip1 | 5 | Polytopic | 0,29 | 2,62E-02 |
| Protocadherin 9 | Pcdh9 | 2 | Type-II | 0,29 | 3,15E-02 |
| Protocadherin-20 | Pcdh20 | 2 | Type-I | 0,30 | 3,28E-02 |
| Plexin-B2 | Plxnb2 | 4 | Type-I | 0,30 | 3,64E-02 |
| Semaphorin-4B | Sema4b | 4 | Type-I | 0,32 | 3,98E-02 |
| Neuronal cell adhesion molecule | **Nrcam** | 4 | Type-I | 0,33 | 4,91E-03 |
| RGM domain family member B | Rgmb | 3 | Type-I | 0,35 | 1,06E-02 |
| Chondroitin sulfate synthase 2 | Chpf | 2 | Type-I | 0,35 | 7,63E-03 |

*Table 1 continued on next page*

Kuhn *et al.* eLife 2016;5:e12748. DOI: 10.7554/eLife.12748

*Table 1 continued*

| Protein Name | Gene Symbol | Peptides | Topology | MEAN (A10KO/CON) | TTEST |
|---|---|---|---|---|---|
| Neogenin | **Neo1** | 27 | Type-I | 0,35 | 3,27E-03 |
| Collagen alpha-1(XI) chain | Col11a1 | 4 | GPI | 0,36 | 4,53E-02 |
| Beta-1,3-galactosyltransferase 6 | B3galt6 | 2 | Type-II | 0,38 | 4,85E-02 |
| Neuroligin-3 | Nlgn3 | 28 | Type-I | 0,38 | 2,97E-04 |

shedding of PTPRU, PTPRG, PTPRK, PTPRS, PTPRT and PTPRD was reduced down to 12%, 17%, 26%, 40% and 43% and 42% respectively while shedding of NLGN1, NLGN3 and NLGN4 was reduced down to 17%, 38% and 56% in the Neuroligin family. Interestingly, NLGN2 shedding was not significantly affected upon ADAM10 knockout. Presynaptic Neurexins 2 and 3, binding partners of Neuroligins, showed only a mild reduction in shedding upon ADAM10 deletion (*Figure 1F*) suggesting that ADAM10 plays only a minor role in their proteolytic processing. Besides these families ectodomain cleavage of member 1 and 3 of the SLITRK family was strongly reduced (*Figure 1F*). Ectodomain cleavage of Protocadherin 8 and 9, which have been proposed to play a role in synaptogenesis, was also reduced (*Yasuda et al., 2007*).

Axon targeting is another important physiological process, which contributes to proper function of the brain. The L1 family, which consists of NgCAM-related cell adhesion molecule (NRCAM), L1 cell adhesion molecule (L1), Close homologue to L1 (CHL1) and Neurofascin has been proposed to play a role in axon targeting and function. We observed that shedding of NRCAM was strongly reduced down to 33%. Shedding of CHL1 was mildly reduced down to 60%. L1 shedding was reduced down to 64% (*Figure 1F*), which is in agreement with their additional cleavage by the protease BACE1 (*Kuhn et al., 2012*). However, the reduction of L1 shedding did just not reach statistical significance. Another axon targeting molecule whose ectodomain shedding was reduced upon Adam10 deletion was Neogenin (NEO1), which belongs to the DCC family. Adam10 deletion also resulted in reduced ectodomain cleavage of CADM1, 3 and 4 which are members of the Cellular adhesion molecule family that has been proposed to play a role in axon myelination (*Maurel et al., 2007*; *Park et al., 2008*). Interestingly, we identified the Low density lipoprotein receptor (LDLR) as a substrate of ADAM10 whose ectodomain shedding was almost completely abolished. Finally, our data revealed that ectodomain shedding of the GPI-anchored membrane-tethered matrix metalloprotease 4 (MT4MMP/MMP17) depends to a great extent on ADAM10 activity.

## Validation of selected substrate candidates by immunoblots

We validated selected ADAM10 substrate candidates with immunoblots. We focused on ADAM10 substrate candidates (marked with yellow rings in *Figure 1C*), where antibodies were available to detect the reduction of the shed ectodomain in the conditioned medium of ADAM10-/- neurons as this allowed a quantitative comparison to the proteomic SPECS analysis. Furthermore, we analyzed full length protein levels to exclude that reduced ectodomain release simply resulted from a reduced full length protein expression in Adam10 knockout neurons.

First, we verified successful Adam10 deletion upon iCre expression by immunoblot (*Figure 2A*, ADAM10). Cre infection strongly reduced levels of immature and mature ADAM10 levels in the neuronal cell lysate. Significantly reduced or almost abolished ectodomain levels confirmed ADAM10 cleavage of the previously described ADAM10 substrates Neuroligin-1 (NLGN1) and N-Cadherin (CDH2) and matched the results in the SPECS experiment (*Figure 2B*), while their full length levels were unchanged or slightly increased (*Suzuki et al., 2012*; *Reiss et al., 2005*). The following novel ADAM10 substrate candidates were validated by immunoblots (*Figure 2A*): LDLR, MT4MMP, LRRC4B, NRCAM, NEO1 and CNTN2. For all proteins ectodomain levels were reduced, while full-length protein levels in the lysate were either unchanged or increased. Importantly, the quantitative reductions of the ectodomain levels measured by immunoblots corresponded very well to the reductions measured by SPECS (*Figure 2B*), demonstrating the quantitative accuracy of SPECS.

Another protein, which was validated is L1 (*Figure 2B*). In ADAM10-/- neurons L1 shedding was reduced by about 30%, both by SPECS and immunoblot (*Figure 2B*), demonstrating that in neurons

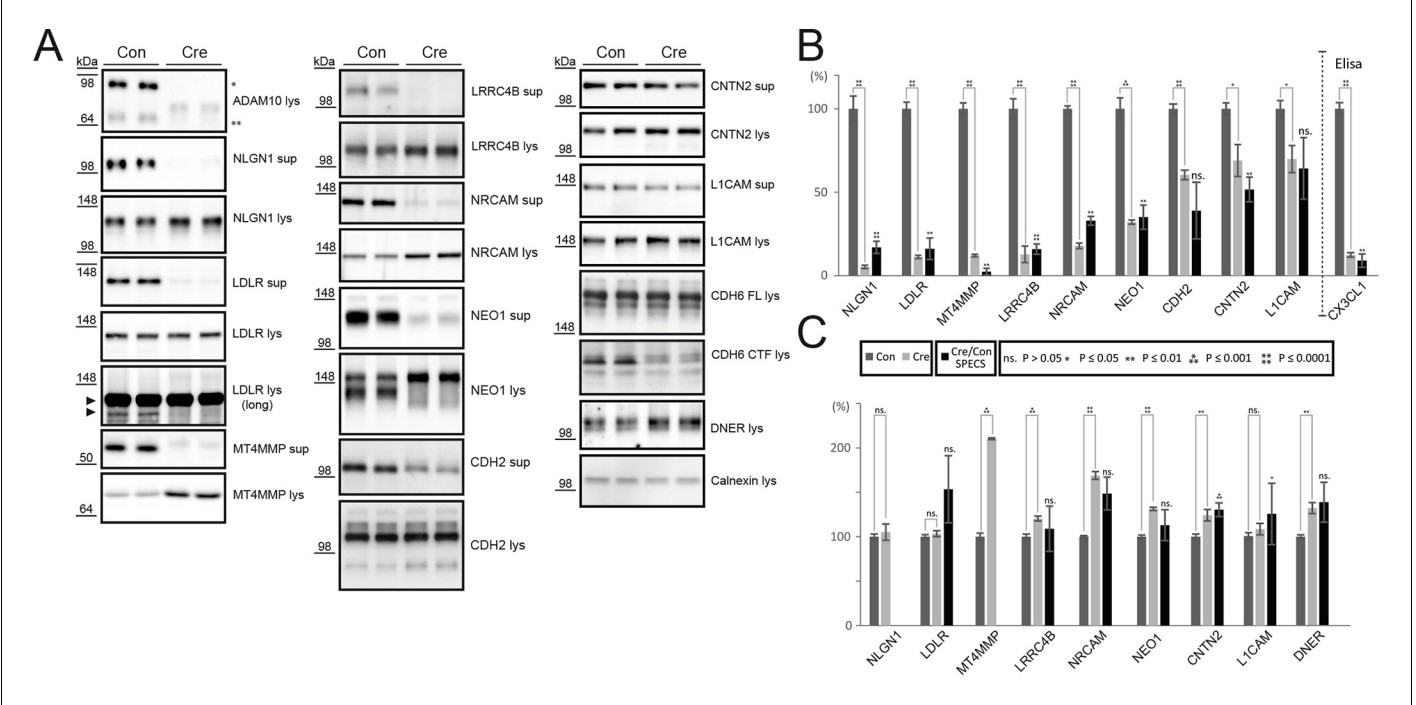

**Figure 2.** Validation of ADAM10 substrates by immunoblot or ELISA. (**A**) Western blots of supernatants (sup, conditioned for 48 hrs) and cell lysates (lys) of neurons expressing endogenous levels of ADAM10 (Con) or no active ADAM10 upon Cre-induced (Cre) ADAM10 knockout. ADAM10 blot shows absent expression of ADAM10 in Cre-transduced neurons. Immunoblots are representative examples from n = 6 experiments. (**B**) Quantification of all biological replicates of representative Western blots in A and comparison to the quantified relative intensity values of remaining shedding in the SPECS experiment: Depicted is the mean of substrate ectodomain levels detected by immunblots of the supernatant under control (CON) and *Adam10* knockout condition by Cre infection (Cre) and the corresponding standard error of the mean (SEM) of 6 biological replicates and p-value calculated with a two-tailed, unpaired t-test. Additionally, we depicted the quantified reduction of ectodomain shedding including SEM and p-value of the SPECS experiment. For CX3CL1 a sandwich ELISA was used to quantify the shedding products in the supernatant. (**C**) Quantification of the mean change in the cell lysate upon ADAM10 deletion of 6 biological replicates including the standard error of the mean and comparison to the SPECS quantified changes in the cellular glycoproteome of 5 biological replicates. For proteins that were also detected in the proteomic analysis of the cellular membrane proteins (*Figure 3*), the proteomic (SPECS) data are also indicated in the graph.

L1 is predominantly cleaved by a protease different from ADAM10. This is in line with previous reports demonstrating that L1 is mostly cleaved by BACE1 in neurons (*Kuhn et al., 2012*; *Zhou et al., 2012*), whereas it is mostly cleaved by ADAM10 in non-neuronal tissues and cell lines (*Riedle et al., 2009*; *Gavert et al., 2007*; *Maretzky et al., 2005*).

For Cadherin-6 (CDH6) no antibody was available against the ectodomain, but an antibody against the cytoplasmic C-terminus. Given that ADAM10 cleavage of a membrane protein does not only lead to the secreted ectodomain, but also to the generation of a C-terminal, membrane bound fragment (CTF), we detected the CDH6 CTF as a read-out of CDH6 cleavage by ADAM10. Hence, we used an antibody directed against the CDH6 cytoplasmic domain and thus detected CDH6 full-length protein and CDH6 C-terminal fragment (CTF) resulting from proteolytic cleavage in the cell lysate. Adam10 deletion strongly reduced CDH6 CTF formation with CDH6 full-length protein levels being unaltered (*Figure 2A*, CDH6), indicating that ADAM10 is required for CDH6 proteolysis. In case of the Notch ligand Delta-Notch EGF receptor (DNER) we observed increased full-length protein levels in the cell lysate, which indicates that DNER is cleaved by ADAM10. However, the DNER ectodomain was not detectable in directly loaded conditioned media of neurons. In case of fractalkine (CX3CL1) we confirmed its known cleavage by ADAM10 (*Hundhausen et al., 2003*) using a sandwich ELISA to detect soluble fractalkine in conditioned media of neurons. Upon Adam10 deletion, we observed a strong reduction of fractalkine ectodomain levels, which corresponded to our prior quantified levels with the SPECS method (*Figure 2B*, CX3CL1).

Taken together, all membrane proteins analyzed by immunoblot could be validated as ADAM10 substrates. When we compared full length protein levels of all investigated proteins to the full length protein levels quantified by mass spectrometry after their SPECS mediated enrichment described below (*Figure 3*), we were able to detect for selected substrates the same increase in the mass spectrometry read out as observed in Western blot (*Figure 2C*).

## Changes in membrane-bound glycoproteins upon ADAM10-deficiency

For the ADAM10 substrates investigated above, reduced shedding in ADAM10-/- neurons was accompanied by unchanged or slightly increased full-length protein levels. For other substrate candidates identified by SPECS no antibodies were available for validation by immunoblot. Thus, to be independent of antibodies we complemented our SPECS analysis of the secretome by a similar analysis of the levels of membrane proteins in the neuronal membrane, which may provide additional validation of more ADAM10 substrate candidates. An added value of this analysis is the opportunity to identify membrane proteins that are not direct ADAM10 substrates, but whose protein levels are instead indirectly changed. For example, it is conceivable that an ADAM10 substrate is part of a protein complex. Hence, its increased full-length protein levels upon ADAM10 deficiency might indirectly stabilize the other complex partner, such as ion channels or neurotransmitter receptors.

Identical to the SPECS method intact neurons were labeled for 48 hr with ManNAz. Instead of collecting the supernatant, labeled cells were then reacted with the biotinylated alkyne in order to label the cellular glycoproteins carrying the modified sugar (*Figure 3A*). Because sialic acid – to which ManNAz is converted - occurs as a terminal glycan modification in the Golgi, our approach is not expected to label all cellular glycoproteins, but only mature glycoproteins within the Golgi and beyond, including the plasma membrane. As the biotin-reagent is membrane-permeable, SPECS is expected to label glycoproteins not only at the plasma membrane, but also in membranes of the secretory and endocytic pathway. Using SPECS for labeling primary murine neurons, we identified 432 glycoproteins of which 37 were significantly changed upon Adam10 deletion considering a p-value of less than 0.05 based on 4 biological replicates (*Figure 3B, C*, *Table 2*, *Supplementary file 2* – A10 knockout surfaceome dataset). Applying multiple hypothesis testing none of the hits would remain significantly changed. However, we could confirm increased levels of APP, CNTN2 and L1CAM by Western blot (see below, *Figure 4*) indicating that correction for multiple hypothesis testing was too strict. Of the 37 candidate proteins, 5 showed reduced levels, while 32 proteins had increased levels (*Table 2*). 62% of the significantly changed proteins possessed a type-I topology (*Figure 3E*) while 22% were polytopic proteins (*Figure 3D*). Among the significantly changed proteins we found previously known substrates like APP, L1CAM or CX3CL1 (*Figure 3E*) (*Lammich et al., 1999*; *Kuhn et al., 2010*; *Hundhausen et al., 2003*; *Colombo et al., 2013*; *Maretzky et al., 2005*). Furthermore, we found substrate candidates that we had already identified in the secretome of ADAM10 knockout neurons like ISLR2, CNTN2 or Neurexin 3 (*Figure 2B, C*). NRCAM which showed increased levels in neurons after ADAM10 knockout in immunoblots (*Figure 2A*) was borderline significant in the surface analysis. The small overlap of only 13 proteins between significantly changed glycoproteins in the secretome and significantly changed glycoproteins in neurons upon ADAM10 knockout showed that ADAM10 cleavage regulated full-length protein levels only in a fraction of cases. Potentially, ADAM10 regulates substrate function by cleavage mainly at the cell surface without affecting total full-length substrate levels, which instead may be subject to lysosomal degradation.

Thus, for the majority of candidate ADAM10 substrates – identified in the secretome analysis – the full-length cellular levels were neither significantly increased nor reduced upon ADAM10 deletion which was also reported previously for the ADAM10 substrates Neuroglin-1 and N-cadherin (*Prox et al., 2013*; *Suzuki et al., 2012*). We observed a mild but statistically significant accumulation in the full-length protein levels of the presynaptically localized Neurexin family member NRXN3 which matches the observed concomitant mild modulation of shedding in the secretome. On the contrary, full-length protein levels of postsynaptic Neurexin binding partners Neuroligin 1, 3 and 4 (NLGN-1/-3/-4) were not increased upon Adam10 deletion. Besides accumulation of full length protein levels of ADAM10 substrates, we additionally detected an accumulation of polytopic membrane proteins like the ionotropic glutamate receptor GLUR1 that plays a role in excitatory synaptic transmission and Reticulon-4 (RTN4) which has been implicated to play a role in axon outgrowth (*Schmandke et al., 2014*). Furthermore, we detected membrane proteins, which accumulate in the

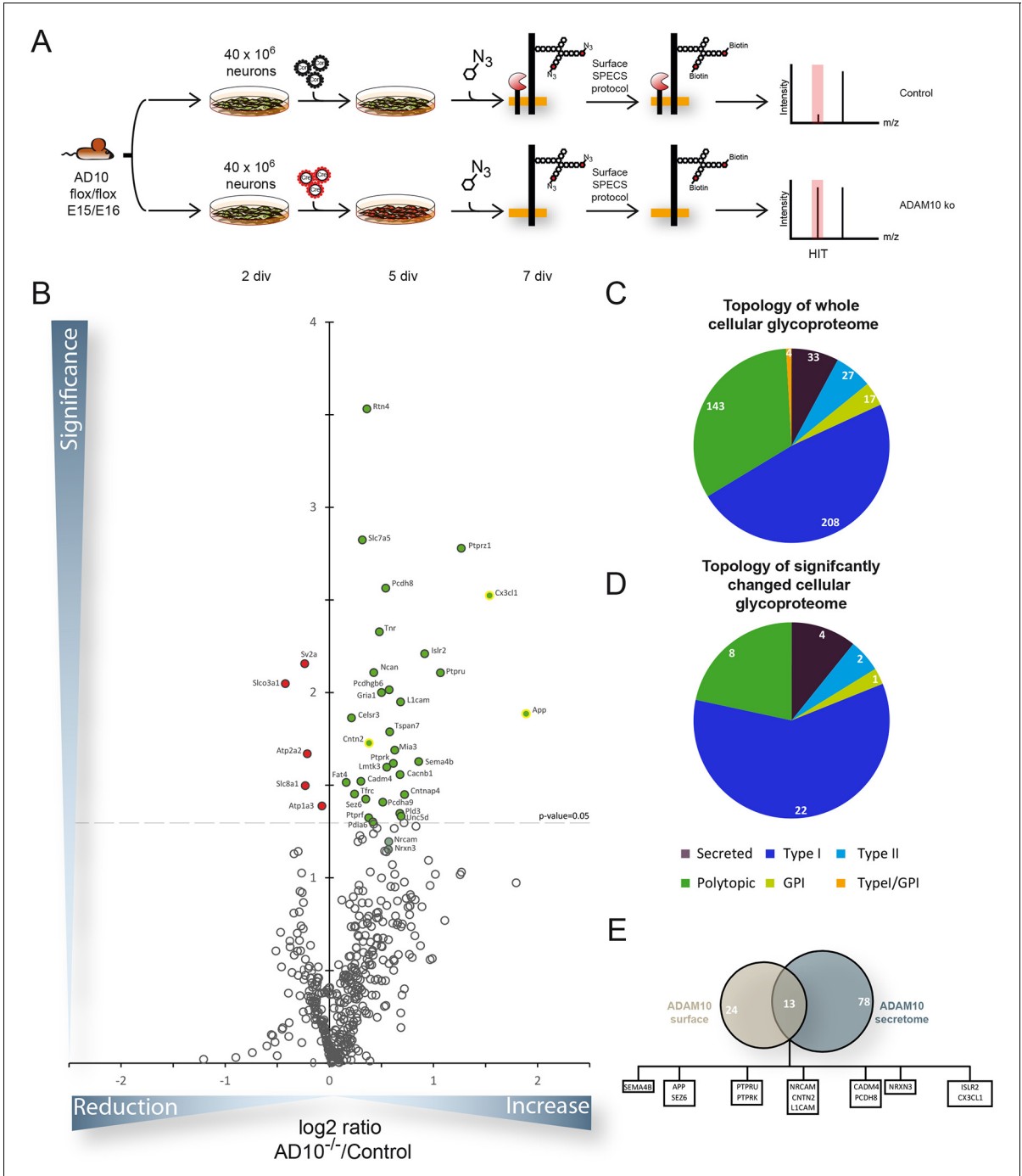

**Figure 3.** Quantitative analysis of glycoproteins in the neuronal cell lysate of conditional Adam10-/- neurons. (**A**) Workflow for the identification of alterations in the cellular glycoproteome upon *Adam10* deletion in primary cortical neurons comprising plating, lentiviral infection, metabolic glycan labeling, biotinylation of glycoproteins on intact neurons and affinity purification protocol and finally measurement via mass spectrometry. (**B**) Volcano plot of the quantitative comparison between the cellular glycoproteomes of wild type (wt) and *Adam10* knockout neurons of all glycoproteins identified in 4 out of 4 experiments in the cellular glycoproteome of *Adam10-/-* and wt neurons. The p-value is depicted as negative decadic logarithm while the fold-change (Adam10-/-/wt) is depicted as log2 value. Significantly changed protein: p≤0.05 (-log10≥1.3). Additionally, we corrected for multiple hypothesis testing with Benjamini Hochberg correction which gave a significance cut-off of p≤0.02. Substrates that are significantly reduced are marked red. Proteins which were validated with immunoblot have a yellow ring. (**C**) Topology of all in four experiments identified glycoproteins in the cellular glycoproteome of *Adam10-/-* and wildtype neurons. (**D**) Topology of identified glycoproteins with significantly increased protein levels in Adam10-/- compared to wt neurons. (**E**) Overlap between proteins significantly reduced in the secretome and proteins significantly increased in the cellular glycoproteome.

**Table 2.** Proteins that are significantly changed in the cellular glycoproteome upon Cre recombinase induced deletion of ADAM10. The table contains all significantly changed proteins upon ADAM10 deletion. Indicated are the names of the proteins, the gene name, number of unique peptides, the mean of the ratio between neurons devoid of ADAM10 and neurons expressing endogenous levels of ADAM10 of 4 biological replicates and the p-value calculated with a two-sided, heteroscedastic t-test based on the relative label-free quantification ratios (LFQ) for the control and the ADAM10 deletion condition. Gene Symbols of previously published ADAM10 substrates are highlighted in bold. (MEAN) Average change between A10 deleted (A10KO) and Control (CON) cells of 4 biological replicates, (SEM) Standard error of the mean. (Peptides) Number of Peptides identified for every protein.

| Protein Name | Gene Symbol | Peptides | Topology | MEAN (A10KO/CON) | TTEST |
|---|---|---|---|---|---|
| Isoform APP695 of Amyloid beta A4 protein | **App** | 15 | Type I | 3,71 | 1,30E-02 |
| Fractalkine | **Cx3cl1** | 6 | Type I | 2,91 | 3,00E-03 |
| Protein Ptprz1 | Ptprz1 | 25 | Type I | 2,41 | 1,66E-03 |
| Receptor-type tyrosine-protein phosphatase U | Ptpru | 7 | Type I | 2,10 | 7,82E-03 |
| Immunoglobulin superfamily containing leucine-rich repeat protein 2 | Islr2 | 16 | Type I | 1,89 | 6,18E-03 |
| Semaphorin-4B | Sema4b | 4 | Type I | 1,81 | 2,36E-02 |
| Contactin-associated protein-like 4 | Cntnap4 | 17 | Type I | 1,65 | 3,55E-02 |
| Fibronectin leucine rich transmembrane protein 3 | Flrt3 | 16 | Type I | 1,64 | 5,05E-02 |
| Netrin receptor UNC5D | Unc5d | 5 | Type I | 1,61 | 4,66E-02 |
| Neural cell adhesion molecule L1 | L1cam | 33 | Type I | 1,61 | 1,12E-02 |
| Voltage-dependent L-type calcium channel subunit beta-1 | Cacnb1 | 9 | Type I | 1,60 | 2,78E-02 |
| Phospholipase D3 | Pld3 | 3 | Type II | 1,60 | 4,49E-02 |
| Melanoma inhibitory activity protein 3 | Mia3 | 23 | Type I | 1,55 | 2,04E-02 |
| Receptor-type tyrosine-protein phosphatase kappa | Ptprk | 12 | Type I | 1,53 | 2,41E-02 |
| Tetraspanin-7 | Tspan7 | 5 | Polytopic | 1,49 | 1,63E-02 |
| Protein Pcdhgb6 | Pcdhgb6 | 9 | Type I | 1,49 | 9,66E-03 |
| Serine/threonine-protein kinase LMTK3 | Lmtk3 | 19 | Type I | 1,47 | 2,53E-02 |
| Protocadherin-8 | Pcdh8 | 32 | Type I | 1,46 | 2,73E-03 |
| Protein Pcdha9 | Pcdha9 | 12 | Type I | 1,43 | 3,90E-02 |
| Glutamate receptor 1 | Gria1 | 21 | Type I | 1,42 | 1,00E-02 |
| Tenascin-R | Tnr | 17 | Secreted | 1,40 | 4,69E-03 |
| Neurocan core protein | Ncan | 6 | Secreted | 1,35 | 7,81E-03 |
| Protein disulfide-isomerase A6 | Pdia6 | 7 | Secreted | 1,34 | 5,00E-02 |
| Contactin-2 | Cntn2 | 18 | GPI | 1,30 | 1,87E-02 |
| Receptor-type tyrosine-protein phosphatase F (Fragment) | Ptprf | 19 | Type I | 1,30 | 4,74E-02 |
| Reticulon-4 | Rtn4 | 39 | Polytopic | 1,28 | 2,94E-04 |
| Isoform 2 of Seizure protein 6 | Sez6 | 6 | Type I | 1,27 | 3,76E-02 |
| Large neutral amino acids transporter small subunit 1 | Slc7a5 | 9 | Polytopic | 1,25 | 1,50E-03 |
| Cell adhesion molecule 4 | Cadm4 | 10 | Type I | 1,23 | 3,02E-02 |
| Transferrin receptor protein 1 | Tfrc | 22 | Type II | 1,18 | 3,52E-02 |
| Cadherin EGF LAG seven-pass G-type receptor 3 | Celsr3 | 35 | Type I | 1,16 | 1,37E-02 |
| Protocadherin Fat 4 | Fat4 | 48 | Type I | 1,12 | 3,05E-02 |
| Sodium/potassium-transporting ATPase subunit alpha-3 | Atp1a3 | 62 | Polytopic | 0,95 | 4,09E-02 |
| Sarcoplasmic/endoplasmic reticulum calcium ATPase 2 | Atp2a2 | 52 | Polytopic | 0,86 | 2,14E-02 |
| Slc8a1 protein | Slc8a1 | 24 | Polytopic | 0,85 | 3,18E-02 |
| Synaptic vesicle glycoprotein 2A | Sv2a | 15 | Polytopic | 0,85 | 6,99E-03 |
| Solute carrier organic anion transporter family member 3A1 | Slco3a1 | 4 | Polytopic | 0,75 | 8,95E-03 |

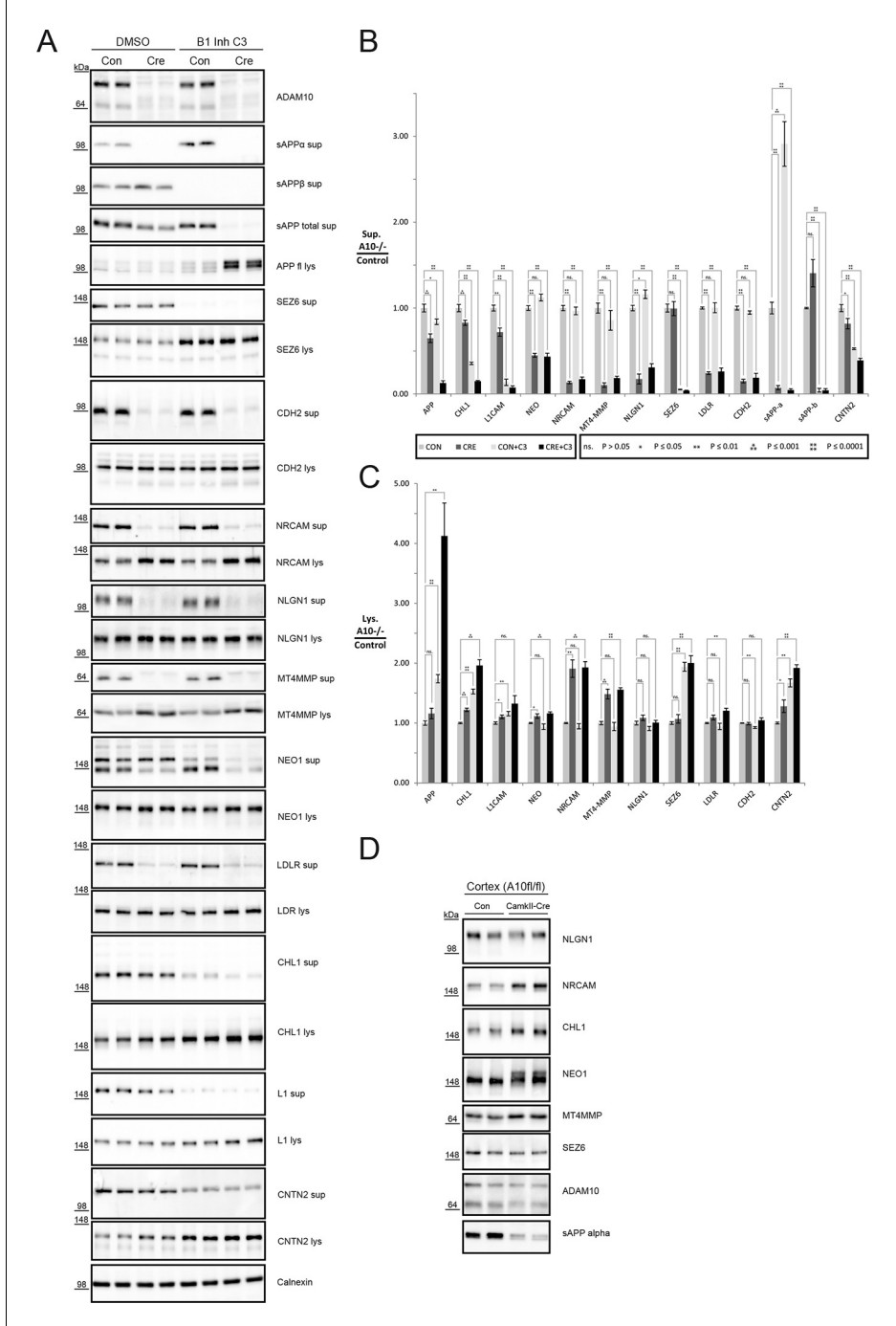

**Figure 4.** Analysis of proteolytic cross talk of ADAM10 and BACE1 for selected substrates. (**A**) Representative Western blots of 48 hr conditioned supernatants and cell lysates of ADAM10 fl/fl neurons treated with or without C3 to inhibit BACE1 and infected with a control virus (Con) or Cre recombinase virus (Cre) to delete *Adam10*. A representative ADAM10 blot shows abolished expression of immature and mature ADAM10. (**B**) Quantification of the mean reduction in ectodomain shedding in the conditioned medium (sup: supernatant), the respective standard error of the mean (SEM) and the significance of six biological replicates calculated with a two-tailed heteroscedastic t-test. A10: ADAM10. (**C**) Quantification of the mean increase of selected substrates in the cellular glycoproteome (lys: lysate), the respective standard error of the mean (SEM) and the significance of 6 biological replicates calculated with a two-tailed heteroscedastic t-test. (**D**) Brain membrane immunoblots from *Adam10*fl/fl and CamkII-Cre *Adam10*fl/fl mice which have lost ADAM10 expression in excitatory neurons, but not in other neurons and non-neuronal cells.

cellular glycoproteome but were not detected in the secretome like LRRTM1 that has been implicated to play a role in modulating the synapse architecture (*Soler-Llavina et al., 2013*). Hence, changes in shedding of a given protein do not necessarily result in accumulation of its precursor in the membrane and, conversely, accumulation of a given protein can also be caused indirectly.

## Cross-talk between ADAM10 and BACE1

Some of the significantly changed proteins in the secretome as well as in the cellular glycoproteome analysis were previously also identified as BACE1 substrates, such as L1, CHL1 and contactin-2. This suggests a potential cross-talk between both proteases and raises the possibility of redundancy between ADAM10 and BACE1 for the cleavage of some of its substrates. To test this possibility, we used immunoblots and analyzed cellular full-length protein levels as well as ectodomain levels in response to BACE1 inhibition, Adam10 deletion or a combination of both (*Figure 4A*). We investigated ectodomain cleavage and cellular levels of 11 substrates (CHL1, L1, CNTN2, NLGN1, NRCAM, LDLR, CDH2, MT4MMP, SEZ6, APP, NEO1), which we selected according to their quantitative reduction in ectodomain shedding in the current ADAM10 and the previous BACE1 SPECS study (*Kuhn et al., 2012*). We selected substrates that were predominantly (NLGN1, NRCAM, MT4-MMP) or partially (CDH2, NEO1) cleaved by ADAM10 or that are predominantly cleaved (SEZ6) and partially cleaved by BACE1 (APP, CHL1, L1, CNTN2). Blotting for ADAM10 confirmed deletion of Adam10 upon lentivirus-mediated expression of Cre recombinase (*Figure 4A*). This was further corroborated by abolishment of sAPPα in ADAM10 deleted neurons. BACE1 inhibition was monitored with the abolishment of sAPPβ (*Figure 4A, B*) and led to a compensatory increase in sAPPα which previously has been described (*Colombo et al., 2013*).

SEZ6 ectodomain cleavage was almost abolished and its cellular levels were increased exclusively upon BACE1 inhibition with the validated inhibitor C3, which shows that SEZ6 ectodomain cleavage is predominantly cleaved by BACE1, in agreement with a previous study (*Kuhn et al., 2012*). However, CDH2, NRCAM, NLGN1, MT4MMP, NEO1 and LDLR ectodomain levels were reduced and their full length protein levels in some cases increased exclusively upon Adam10 deletion while simultaneous BACE1 inhibition had no additional effect. In contrast to the L1 family member NRCAM, Adam10 deletion and BACE1 inhibition had an additive effect in the reduction of ectodomain cleavage and accumulation of full length protein levels for the members CHL1 and L1CAM in a similar fashion as for APP (*Figure 4A, B*). This also held true for CNTN2. However, in case of APP the lack of BACE1 cleavage seems to be compensated by ADAM10, which does not appear to be the case for CHL1, L1CAM or CNTN2. In summary, the 11 investigated substrates can be subdivided into three classes, namely predominant ADAM10 substrates, predominant BACE1 substrates and substrates which are cleaved by ADAM10 and BACE1, demonstrating that there is a cross-talk between ADAM10 and BACE1 for some but not all substrates.

Finally, we tested whether the substrates identified in neurons would also be cleaved by ADAM10 in vivo. Therefore, we analyzed the brain membrane fraction of conditional Adam10 knockout mice that had been crossed with a postnatal neuron-specific CamKII-Cre driver line at P20 (*Prox et al., 2013*). Due to technical limitations we were only able to analyze the membrane fraction of ADAM10 knockout brains. We observed a 50% reduction of total ADAM10 in the brain membrane fraction, which is in line with the excitatory neuron-specific CamkII-Cre driver line sparing GABAergic neurons and the fact that ADAM10 besides neurons is also expressed in glial cells. Similar to our in vitroexperiments with primary cortical neurons full-length levels of NLGN1 were not changed in brain membrane extracts. In contrast to NLGN1, we observed a clear increase of full-length protein levels for NEO1, NRCAM, CHL1 and MT4-MMP (*Figure 4D*) similar to what we had observed previously in neuronal lysates upon Adam10 deletion (*Figures 2A,4A*). As expected, the predominant BACE1 substrate SEZ6 showed no change upon Adam10 deletion (*Figure 4D*).

## Altered connectivity in olfactory bulb of Adam10$^{-/-}$ mice

Several of the newly identified ADAM10 substrates, including the newly identified substrates NRCAM and CHL1, have functions in axon targeting (*Demyanenko et al., 2011*; *Heyden et al., 2008*; *Montag-Sallaz et al., 2002*). Thus, it appears possible that Adam10-/- mice present defects in brain connectivity. To test this we analyzed P20 conditional Adam10fl/fl knockout mice that had been crossed with a postnatal neuron-specific CamKII-Cre driver line, resulting in forebrain specific

Adam10 deletion in excitatory neurons, while the cerebellum is not affected (*Prox et al., 2013*; *Casanova et al., 2001*). Brain areas, where connectivity changes can be well studied, are the olfactory bulb and the hippocampus. Staining of olfactory glomeruli in the olfactory bulb with the plant lectin DBA revealed 40% of diffuse olfactory glomeruli in Adam10-/- mice compared to 10% diffuse olfactory glomeruli in control Adam10fl/fl mice. Additionally, individual axons (white arrows) that seemed to project to two glomeruli, were detected in the ADAM10-/-, but not in control mouse olfactory bulbs (*Figure 5A*). Such mistargeted projections have previously been observed in mice lacking our newly identified ADAM10 substrate NRCAM or lacking the BACE1 substrate CHL1 and may result from mistargeted axons that have not been correctly eliminated during development (*Heyden et al., 2008*).

Staining in the hippocampus for the synaptic marker protein synatophysin revealed additional changes in ADAM10-/- mice. Synaptophysin stains synaptic terminals of mossy fibers, which project from the dentate gyrus to the dendritic arbor of hippocampal pyramidal cells in CA3 (*Figure 5C*). In control mice the pyramidal somata were not stained and clearly separated from synaptophysin staining in the stratum lucidum. This separation was lost in Adam10-/- mice, where a significantly increased number and area of synaptophysin positive mossy fiber terminals in the pyramidal layer revealed aberrant mossy fiber projections on hippocampal pyramidal somata in CA3 (*Figure 5D*). Interestingly, this phenotype is also observed in mice lacking the ADAM10 substrate NCAM (*Cremer et al., 1998*) and in mice lacking CHL1 (*Montag-Sallaz et al., 2002*), which is a substrate for both ADAM10 and BACE1 (*Heyden et al., 2008*). Taken together, these results reveal mistargeted axons in the hippocampus and olfactory bulb of Adam10-/- mice which may result from impaired processing of NRCAM, CHL1 and NCAM.

Other ADAM10 substrates, including APP and NLGN1, have physiological roles in brain development, for example in synapse formation/plasticity (*Ring et al., 2007*; *Weyer et al., 2014*; *Suzuki et al., 2012*; *Heyden et al., 2008*; *Montag-Sallaz et al., 2002*; *Brennaman et al., 2013*; *Hick et al., 2015*). In fact, one of the previously reported phenotypes in ADAM10-/- brains are synaptic alterations. In the stratum radiatum of hippocampal region CA1 in ADAM10-deficient but not in wild-type mice, enlarged stubby dendritic spines filled with mitochondria were observed (*Prox et al., 2013*). Interestingly, APP also has a role in synapse formation and maintenance, suggesting that loss of the ADAM10 cleavage product sAPPα in ADAM10-/- mice may be responsible for their synaptic alterations. To this end we crossed ADAM10-/- mice with sAPPα knockin (sAPPαki) mice that do no longer express APP-FL but solely the secreted APPsα ectodomain due to a stop codon insertion into the mouse APP locus behind the α-secretase cleavage site (*Ring et al., 2007*). However, knock-in of sAPPα in ADAM10 -/- mice, confirmed by Western blot analysis (*Figure 6A*), as well as ultrastructural analysis (*Figure 6B*) failed to rescue this synaptic phenotype, demonstrating that dysregulation of another or even several different ADAM10 substrates cause the deficits in synaptic morphology and function. Additionally, increased astrocytic activation demonstrated by enhanced GFAP expression could not be rescued by sAPPα expression (*Figure 6A*).

## Discussion

Ectodomain shedding of membrane proteins is a fundamental mechanism to control intercellular communication, the interaction between cells and their environment. Our study demonstrates that ADAM10 is a major shedding enzyme in the nervous system and that a loss of ADAM10 leads to defects in neuronal connectivity.

The metalloprotease ADAM10 plays manifold roles in development and disease. Studying ADAM10 substrates in primary cells or tissue has not been possible until the advent of conditional Adam10 knockout mice as constitutive Adam10 knockout mice die latest at embryonic day 9.5 (E9.5). These mice succumb to malformations like a retarded heart development, extracorporeal formation of blood vessels and improper formation of somites, which are considered to result from a lack in proteolytic processing of Notch by ADAM10 (*Hartmann et al., 2002*; *Jorissen et al., 2010*; *Krebs et al., 2000*; *Kageyama and Ohtsuka, 1999*). However, postnatal neuron-specific disruption of Adam10 in the brain leads to impaired learning, defects in long term potentiation and seizures which are considered to result from defects in synapse function and architecture and microglial and astrocytic activation (*Prox et al., 2013*). These phenotypes point to an important postnatal function of ADAM10 via cleavage of additional substrates beyond Notch. So far, Notch and other ADAM10

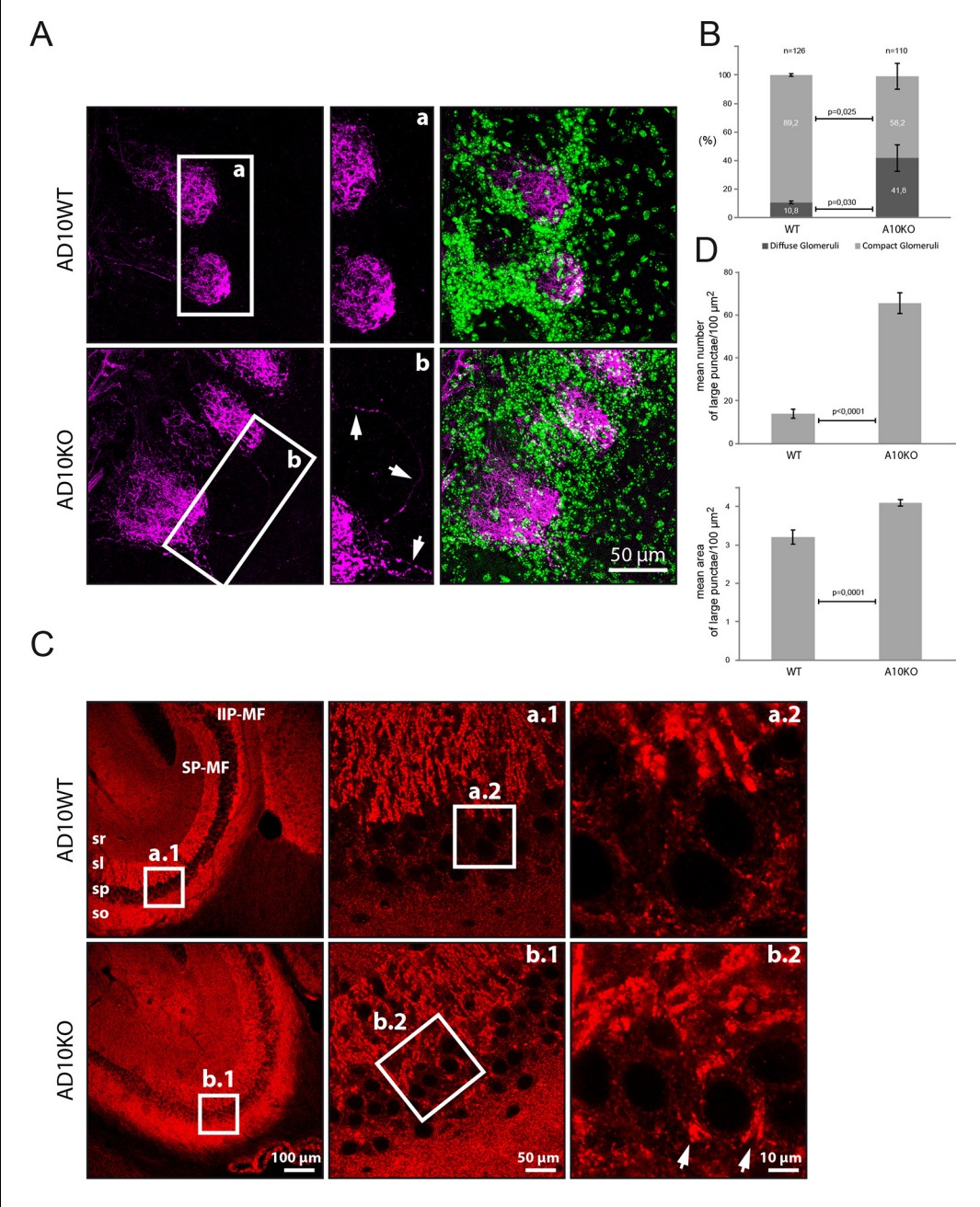

**Figure 5.** Axon targeting and synaptic alterations in ADAM10 deficient mice (**A**) Olfactory nerve axons in adult ADAM10-deficient mice. Confocal microscopy picture of olfactory bulb frontal sections stained for the detection of the olfactory nerve axons labeled with the plant lectin DBA conjugated to biotin (in magenta). Cell nuclei labeled with DAPI appear in green. In wild-type mice (upper panel), olfactory nerve terminals generally project to only one olfactory glomerulus (AD10-fl/fl). In contrast, in ADAM10-deficient mice (AD10-fl/fl+CamkII-Cre) (lower panel), some olfactory axons terminate in two glomeruli or pass through the glomerular layer and terminate in the external plexiform layer. In addition, the arborizations of some olfactory axons extend outside a particular olfactory glomerulus. (**B**) Quantification of olfactory glomeruli morphology in adult ADAM10-deficient mice. In wild-type mice the majority (~90%) of glomeruli stained with the lectin DBA display a compact defined morphology. In ADAM10-deficient mice (AD10-fl/fl+CamkII-Cre), significantly less glomeruli are compact and many glomeruli appear more diffuse. (**C**) Mossy fiber organization in adult ADAM10-deficient mice. Sagittal sections stained for the presence of synaptophysin (red). Confocal microscopy showing the distribution of mossy fiber terminals in the CA3 subfield of the hippocampus. In wild-type mice (AD10-fl/fl) (upper panel) the mossy fibers are organized in the infra-inter (IIP-MF) and supra (SP-MF) pyramidal bundles both terminating in large synaptic boutons on

*Figure 5 continued on next page*

*Figure 5 continued*

pyramidal cell dendrites in the stratum lucidum (sl) of the CA3. CA3 can be clearly distinguished from the stratum pyramidale (sp) containing the pyramidal cell bodies and small synatophysin positive inhibitory synapses. In ADAM10-deficient mice (AD10-fl/fl+CamkII-Cre) (lower panel), mossy fiber terminals are also detected throughout the sp surrounding the pyramidal cell soma. (**D**) Quantitative comparison of mossy fiber terminals in the stratum lucidum with respect to number and size between adult wild type and ADAM10-deficient mice. Large synaptophysin-labeled puncta indicative for mossy fiber synaptic boutons are significantly more frequent in the stratum lucidum of ADAM10-deficient mice (AD10-fl/fl+CamkII-Cre).

substrates like N-Cadherin, fractalkine, Neuroligin-1 or E-Cadherin have already been described (*Hundhausen et al., 2003*; *Suzuki et al., 2012*; *Reiss et al., 2005*; *Maretzky et al., 2005*). However, only few substrates like N-Cadherin, Neuroligin-1, Notch-1 and Notch-2 have been shown to be cleaved in primary cells like neurons due to the aforementioned embryonic lethality of constitutive Adam10 knockout mice and the lack of conditional Adam10 knockout models until recently (*Prox et al., 2013*; *Suzuki et al., 2012*; *Zheng et al., 2012*).

Here, we provide the first systematic analysis of ADAM10 substrates in primary cortical neurons of a conditional Adam10 knockout mouse. Applying SPECS to enrich newly synthesized glycoproteins from conditioned media and the cell surface of primary neurons we identified more than 300 glycoproteins in the secretome, which is slightly more than in our previous study identifying BACE1 substrates in neurons (*Kuhn et al., 2012*). Additionally, we identified and quantified more than 400 membrane glycoproteins on or close to the cell surface of neurons which is 10 times more membrane proteins than in a previous study where cell surface biotinylation was used, but without the SPECS labeling (*Sanz et al., 2015*). This demonstrates the power of the metabolic SPECS labeling to an in-depth quantification of the neuronal membrane proteome. Our analysis revealed that almost 50% of all membrane proteins released into the secretome are reduced upon Adam10 deletion of which 70% possess a type-I topology. Furthermore, we found that 13 secreted proteins like clusterin were reduced. Because ADAM10 cleaves membrane-bound substrates, the reduced clusterin levels are likely to be the consequence of secondary effects such as an increased internalization due to stabilization of its surface receptor in ADAM10-deficient neurons. The large number of substrates makes ADAM10 a major sheddase in the nervous system and an important modifier of the neuronal secretome. Among the candidate ADAM10 substrates in the secretome we could identify whole protein families having roles in synapse function and architecture. Their reduced cleavage upon Adam10 deletion fits to the phenotypes described in neuron-specific conditional Adam10 knockout mice, including defects in long term potentiation and seizures (*Prox et al., 2013*). For example, Neurexins and Neuroligins are known to form trans-synaptic complexes which are especially necessary for synapse function (*Figure 7*) (*Bang and Owczarek, 2013*; *Südhof, 2008*). Adam10 ablation caused a strong reduction in ectodomain shedding of postsynaptic NLGN1, 3 and slightly less pronounced for NLGN4. Hence, shedding of Neuroligins might negatively regulate their synaptic function (*Suzuki et al., 2012*). Surprisingly, NLGN2 shedding was not affected by Adam10 deletion, which might be explained by the distinct roles played by these proteins. In fact, NLGN1 and 3 contribute to the function of excitatory, glutamatergic synapses while NLGN2 contributes to the function of inhibitory, GABAergic synapses (*Kang et al., 2014*) suggesting that ADAM10 might preferably be involved in the modulation of glutamatergic, excitatory but not GABAergic inhibitory synapse functions. This is additionally supported by the fact that the ionotropic glutamate receptor GLUR1 (Gria1), previously described to be modulated by synaptic NLGN1 levels (*Wittenmayer et al., 2009*), was increased in the cellular glycoproteome upon Adam10 deletion while GABA receptors which are known to interact with NLGN2 were not increased (*Kang et al., 2014*). Besides the reduction of NLGN1, 3 and 4 shedding, Adam10 deletion resulted in a concomitant shedding reduction of the NLGN receptors NRXN 2 and 3 while the full length NRXN precursors mildly accumulated in the cellular glycoproteome. We additionally observed reduced ectodomain shedding for almost all members of the receptor tyrosine phosphatase family (PTPR) and the Slit and receptor tyrosine kinase domain (SLITRK) family which both play roles in synapse function and formation (*Figure 7*) (*Um et al., 2014*; *Yim et al., 2013*). Specific members of both families like for example PTPRD and SLITRK3 have been shown to interact with each other transsynaptically (*Um et al.,*

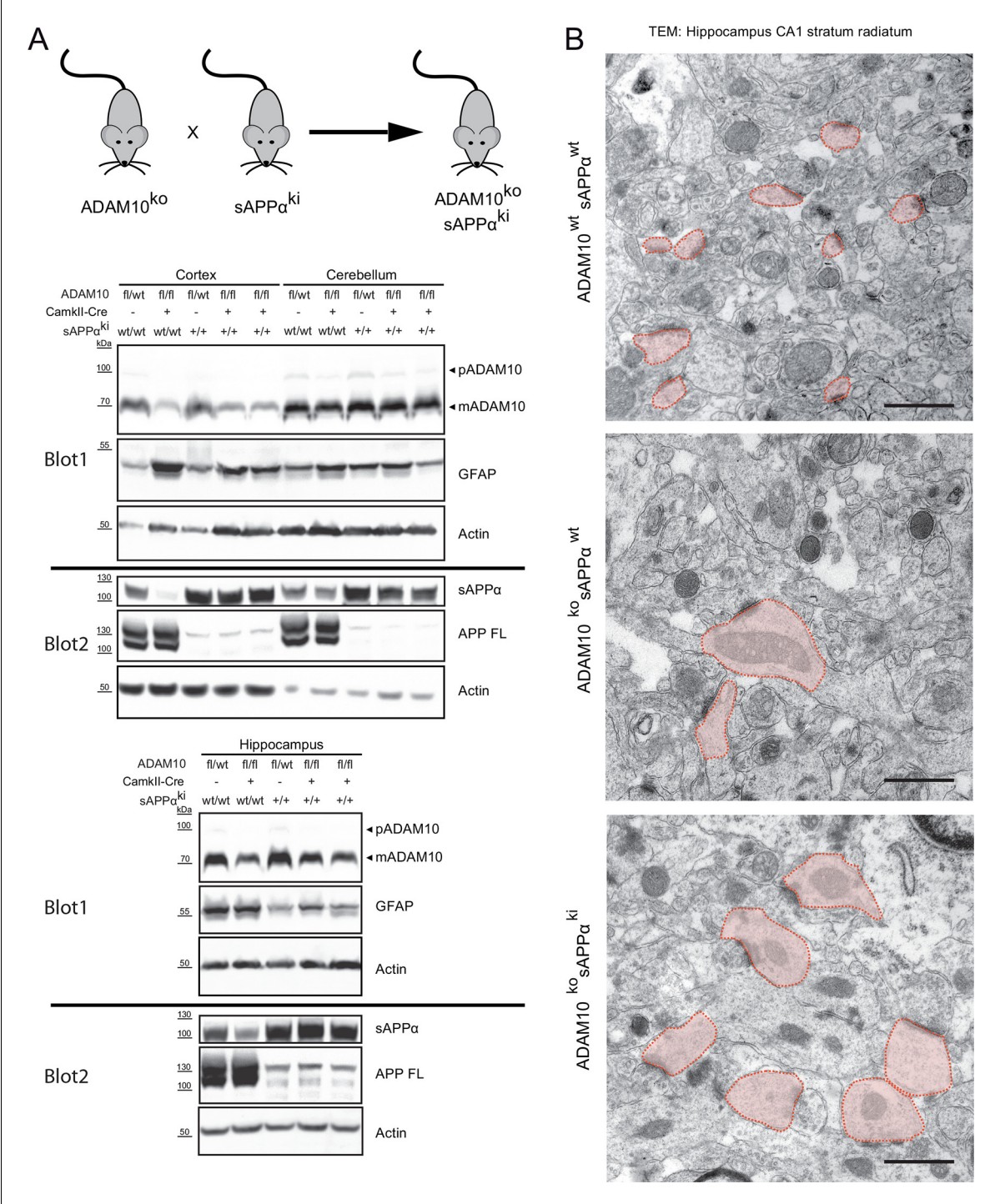

**Figure 6.** Synaptic alterations in ADAM10 deficient mice are not rescued by APPsα overexpression (**A**) Mice deficient for ADAM10 in adult brain (ADAM10-fl/fl+CamkII-Cre) and transgenic for a sAPPα knockin into the APP locus (ADAM10ko sAPPki) thus expressing only sAPPα, were generated by crossing sAPPα knockin mice (sAPPki) and ADAM10-fl/fl+CamkII-Cre knockout mice (ADAM10ko) for two generations. Immunoblot analysis of extracts from cortex, cerebellum and hippocampus from 21 days old mice revealed that ADAM10-expression was clearly reduced in cortex and hippocampus of conditional ADAM10 knockout mice (A10fl/fl, CamkII-Cre pos.). No reduction of ADAM10 was observed in cerebellum since this tissue was not targeted by the CamkII-Cre driver. The proform (pADAM10: 100 kDa) and the mature form (mADAM10: 70kDa) of ADAM10 are depicted. Expression of sAPPα (105 kDa) was detectable in sAPPα transgenic mice (sAPPki +) but absent in wild type mice. Full length APP (APPfl) (105 kDa) expression was lost in sAPPα knockin mice but is present in wild type mice. As reported (**Prox et al., 2013**) GFAP (45 kDa) expression in cortex of 21 days old brains was increased in the ADAM10 knockout mice (ADAM10 ko). This pathology was not reversed upon sAPPα expression (sAPPki +), hinting to an ongoing

*Figure 6 continued on next page*

*Figure 6 continued*

astrogliosis. (B) High resolution electron micrographs of spines in the hippocampal CA1 stratum radiatum from 21 days old mice. Conditional ADAM10 deficiency (ADAM10ko) leads to enlarged and stubby spines (light red shaded). The presence of sAPPα (sAPPki) in ADAM10 knockout mice is not sufficient to rescue the alterations in spine morphology. In wildtype hippocampus spines are characterized as tiny spine heads with no organelles. Scale bars: 1 μm.

*2014*; *Takahashi et al., 2012*). While SLITRK1, 2, 4 and 5 and PTPRS have been implicated in the formation of excitatory synapses, SLITRK3 and PTPRD seem to be involved in the formation of inhibitory synapses (*Yim et al., 2013*).

Impaired cognitive functions can also result from impaired axon targeting and function leading to impaired signal transmission and wrong connection of cortical regions in the brain. We observed mistargeted axons of the olfactory epithelium projecting instead of usually to one olfactory glomerulus to two olfactory glomeruli in the olfactory bulb of ADAM10-deficient mice. Similar axon targeting phenotypes have also been reported upon knock-out of L1 family members Nrcam and Chl1 while L1 knockout mice succumb to the much more severe CRASH syndrome with corpus callosum

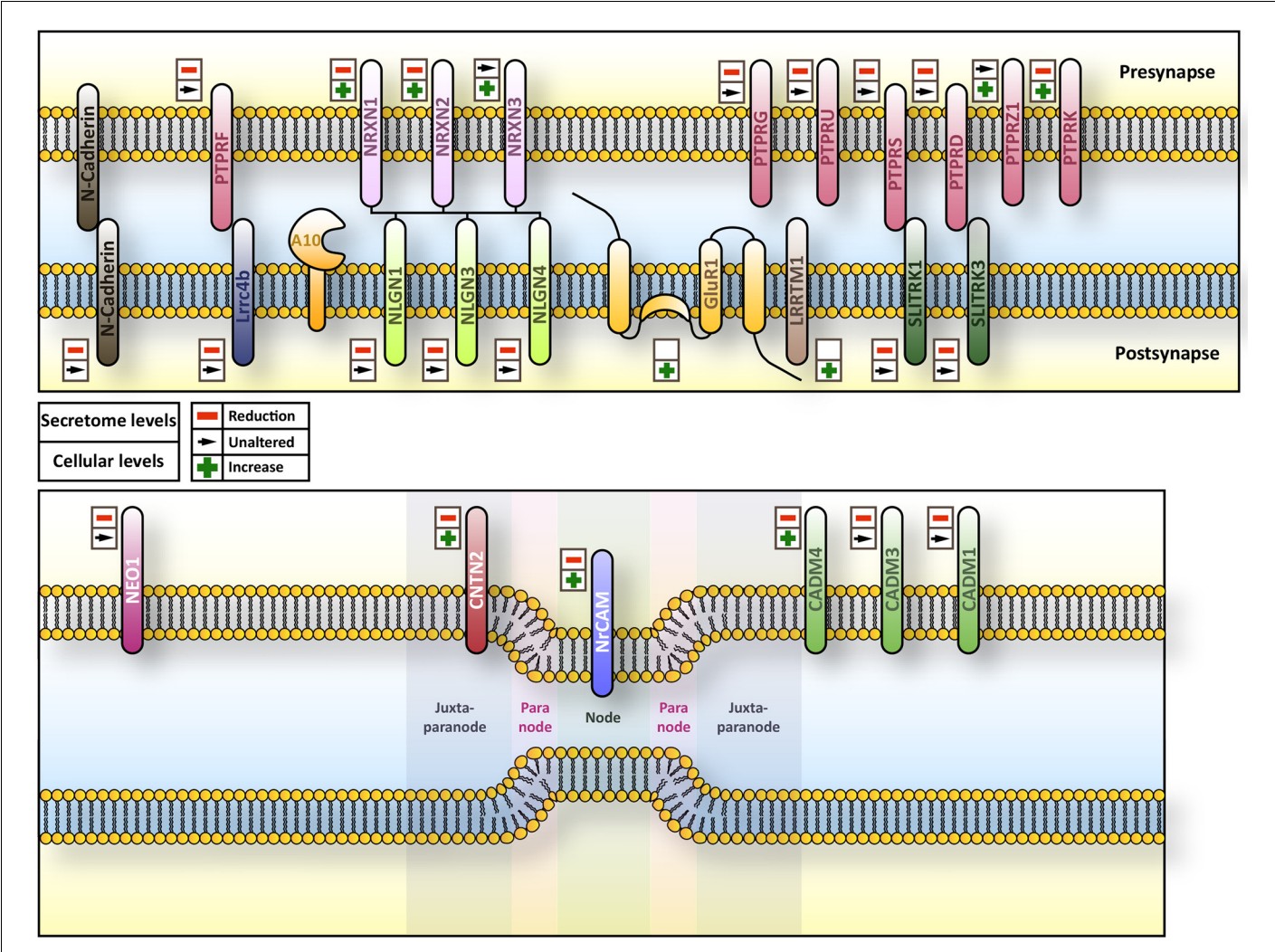

**Figure 7.** Overview of subcellular localization and interaction partners of ADAM10 substrates. Overview of the subcellular localization and interaction partners of selected substrates of ADAM10 and their respective behavior in terms of ectodomain shedding and cellular amount upon Adam10 deletion. Upper symbol refers to ectodomain shedding while lower symbol indicates the cellular levels of the protein.

hypoplasia and mental retardation (*Heyden et al., 2008*; *Montag-Sallaz et al., 2002*; *Demyanenko et al., 2011*; *Kolata et al., 2008*). Indeed, we observed reduced ectodomain shedding of all L1 family members (L1, CHL1, NRCAM and Neurofascin (NFASC)) with NRCAM shedding being reduced the strongest. In contrast, shedding of L1 and CHL1, previously identified as major substrates of BACE1 in neurons (*Kuhn et al., 2012*; *Zhou et al., 2012*) were only mildly modulated upon ADAM10 knockout in line with previous data that demonstrate cleavage of CHL1 and L1 by metalloproteases in non-neuronal cells (*Maretzky et al., 2005*; *Naus et al., 2004*). Simultaneous BACE1 inhibition and Adam10 knockout resulted in a further reduction of ectodomain shedding of CHL1 and L1 in a similar fashion to APP (*Colombo et al., 2013*; *Zhou et al., 2012*). While for L1 differential functional outcomes of ADAM10 and BACE1 cleavage have not been investigated so far, it recently has been shown that CHL1 cleavage by BACE1 is indispensible for Semaphorin 3a induced axon repulsion while inhibition of metalloprotease mediated CHL1 cleavage had no impact which suggests that modulation of axon growth in the axonal or presynaptic compartment via CHL1 is only modulated by BACE1 (*Barão et al., 2015*). Interestingly, as shown here for conditional ADAM10 knockout CamKII-Cre mice, misplaced hippocampal mossy fiber terminals were also observed in CHL1- or NCAM-deficient (*Colombo et al., 2013*; *Yasuda et al., 2007*) but also in BACE1-deficient mice (*Hitt et al., 2012*). Likewise, CHL1-, NrCAM-, BACE1-, or ADAM10-deficiency all result in mistargeted axons in olfactory glomeruli (*Montag-Sallaz et al., 2002*), which may indicate functions for the refinement of connectivity (*Montag-Sallaz et al., 2002*). Nevertheless, other identified substrates, such as neogenin (NEO1) and contactin-2 (CNTN2) may also contribute (*Denaxa et al., 2005*; *Braisted et al., 2000*), which makes a substrate phenotype correlation difficult.

Conditional Adam10 knockout mice also show activated microglia upon postnatal, neuronal deletion of Adam10 (*Prox et al., 2013*). One important chemokine expressed by neurons is fractalkine (CX3CL1) which has previously been proposed to be cleaved by ADAM10 and ADAM17 (*Hundhausen et al., 2003*; *Garton et al., 2001*). Abolished constitutive shedding of CX3CL1 and increased CX3CL1 full length protein levels in neurons upon knockout of ADAM10 demonstrate that ADAM10 regulates CX3CL1 levels in neurons (*Figure 3B*). The biological role of ADAM10 cleaved soluble CX3CL1 might be a reduction of microglial activation leading to neuroprotection. This is supported by the finding that microglial activation and neuronal cell death in the substantia nigra upon MPTP treatment are increased in a CX3CL1 knockout mouse model and can be rescued by soluble but not membrane bound CX3CL1 suggesting a neuroprotective role of soluble CX3CL1 (*Ueno et al., 2013*; *Nash et al., 2013*; *Morganti et al., 2012*).

We also identified low density lipoprotein receptor (LDLR) as a novel substrate of ADAM10. LDLR has been described to undergo induced release upon treatment with the phorbolester PMA and its c-terminal fragment is subject to intramembrane proteolysis by γ-secretase (*Tveten et al., 2013*). Here, we show that constitutive shedding of LDLR is catalyzed by ADAM10. Soluble LDLR has been shown to be able to bind to LDL. However, functional consequences of ADAM10 catalyzed cleavage of LDLR like an altered LDL uptake have not been investigated so far.

Another important finding was that in contrast to other proteases like ADAM17 or ADAM9 (*Kuhn et al., 2010*), cellular levels of the GPI-anchored, neuronally expressed metalloprotease MT4MMP increased while MT4MMP ectodomain shedding was abolished upon Adam10 deletion (*Rikimaru et al., 2007*). MT4MMP is assumed to mostly cleave soluble proteins in the extracellular matrix (*Itoh, 2015*). However, since little is known about the physiological substrate spectrum of MT4MMP in neurons, we cannot exclude that some of our observed changes in the ADAM10-/- neurons and their secretome may be an indirect consequence of the reduced MT4MMP shedding.

Our data revealed that some proteins are exclusive substrates of ADAM10 (NRCAM, CX3CL1) while others are exclusively cleaved by BACE1 (SEZ6, APLP1). Finally, there is a group of proteins that can be cleaved by both proteases (APP, CHL1, L1). This specificity may result from differential sorting of the substrates towards the pre- and postsynaptic compartment, or from primary sequence constraints in the juxtamembrane region. However, this classification will be extended in the future as substrates of other in neuron expressed proteases like MT4MMP and MT5MMP have not been determined in the brain yet (*Rikimaru et al., 2007*; *Jaworski, 2000*).

Taken together, the identification of novel ADAM10 substrates demonstrates that ADAM10 has a large spectrum of neuronal substrates, but the ADAM10 substrate repertoire is presumably even larger. For example Notch1, an established ADAM10 substrate, was not detected in our analysis, potentially because its expression level was below the detection limit or because it was not

expressed in the neurons at the time-point of analysis or because its ADAM10-mediated shedding is ligand-dependent. Additionally, while we identified ADAM10 substrates in neurons, ADAM10 is also expressed in astrocytes and oligodendrocytes and certain proteins are expressed at later developmental stages than what we have investigated (*Sakry et al., 2014*; *Ludwig et al., 2005*). Furthermore, it is possible that certain biological effects of Adam10 deletion cannot be investigated in dissociated in vitro cultures of neurons but require the context of brain tissue to provide the correct extracellular matrix and tissue architecture, which is provided by slice cultures or even better in vivo.

In summary, our study demonstrates that ADAM10 is a major sheddase in the brain and is required for correct axon targeting in the olfactory bulb and the hippocampus of conditional ADAM10-/- mice. The newly identified substrates demonstrate that ADAM10 has a fundamental role in the brain and provide a molecular explanation for cognitive deficits, seizures and axon mistargeting in neuron-specific ADAM10 knockout mice. Finally, these findings will also aid the prediction of potential side effects of ADAM10-activating drugs as they are considered for the treatment of AD and prion disease.

## Materials and methods

### Antibody production in rat

Antibodies against murine CHL1 (clone 7B2, IgG2A) and murine Neogenin (NEO1) (clone 21A8 IgGG1) were produced by fusing murine CHL1 ectodomain (aa26-1006) and murine Neogenin ecto-domain (aa45-1101) to an N-terminal CD5 signal peptide and a C-terminal 2XStrepII tag which subsequently were expressed in HEK293T cells. Recombinant ectodomain was purified from 300 ml conditioned media with 300 µl Streptactin sepharose (IBA GmbH, Göttingen, Germany) according to the instructions of the manufacturer. 50 µg of each purified fusion protein (mCHL1-2XStrepII, mNEO1-2XStrepII) were injected intraperitoneally (i.p.) and subcutaneously (s.c.) into LOU/C rats using incomplete Freund's adjuvant supplemented with 5 nmol CpG 2006 (TIB MOLBIOL, Berlin, Germany). After a six week interval a final boost with 50 µg hAPP and CpG 2006 was given i.p. and s.c. three days before fusion. Fusion of the myeloma cell line P3X63-Ag8.653 with the rat immune spleen cells was performed according to standard procedures. Hybridoma supernatants were tested in a solid-phase immunoassay with mCHL1-2XStrepII, mNEO1-2XStrepII or an irrelevant StrepII fusion protein coated to ELISA plates. Antibodies from tissue culture supernatant bound to mCHL1-2XStrepII or mNEO1-2XStrepII were detected with HRP conjugated mAbs against the rat IgG isotypes (TIB173 IgG2a, TIB174 IgG2b, TIB170 IgG1 all from ATCC, R-2c IgG2c homemade), thus avoiding mAbs of IgM class. HRP was visualized with ready to use TMB (1-StepTM Ultra TMB-ELISA, Thermo, Braunschweig, Germany). MAbs that reacted specifically with mCHL1-2XStrepII or mNEO1-2XStrepII were further analyzed in non-reducing Western blot and immunocytochemistry.

### Antibody production in chicken

Murine DNER ectodomain (aa26-620) was fused to an N-terminal CD5 signal peptide and a C-terminal biotin accepting peptide and 6XHIS tag. This construct was expressed in HEK293T cells with biotin ligase (BirA) fused to an N-terminal CD5 signal peptide and a C-terminal KDEL endoplasmic reticulum (ER) retention motif to target BirA to the ER. Recombinant ectodomain was purified from serum free media with a 1 ml Nickel NTA column (GE Health Care, Solingen, Germany) according to the instructions of the manufacturer. Briefly, the column was equilibrated with 5 column volumes (CV) 5 mM Imidazol in phosphate buffered saline (PBS), 300 ml serum free media containing the recombinant protein and supplemented with 5 mM Imidazol were loaded on the column at a flow rate of 2 ml/min. Afterwards the column was washed with 5 CV 5 mM Imidazol in PBS. Bound DNER-BAP/HIS was eluted with 5 CV 200 mM Imidazol in PBS. LSL chicken hatching eggs were obtained from LSL Rhein-Main (Gut Heinrichsruh, Berglern). Birds were hatched and raised at the Institute for Animal Physiology, Department of Animal Science, LMU Munich. The birds were fed ad libitum with a standard chicken diet. At an age of eight months hens were immunized i.m. with 300 µg purified DNER protein mixed with Freund's adjuvant complete (Sigma-Aldrich, Munich, Germany). Birds were boosted i.m. with 300 µg protein plus Freund's incomplete adjuvant (Sigma-Aldrich) three weeks after the initial immunization. Eggs were collected two weeks after the boost for a period of two weeks. Animal experiments were authorized by the Regierung von Oberbayern (55.2-1-54-2532.6-

12-09). DNER chicken antibodies were purified by antigen affinity purification. 300 µg DNER-BAP-HIS was bound to 300 µl streptavidin bead slurry (50% w/v). The purified immunoglobulin fraction of chicken egg yolk was loaded on the DNER-BAP-HIS Streptavidin column to isolate DNER specific antibodies. Bound DNER antibodies were eluted with acidic elution at pH 2.5.

## Antibodies from commercial or academic sources

The following antibodies were used in this study: ADAM10 (Abcam/Epitomics, Cambridge, UK, EPR5622), Actin (Sigma-Aldrich, A2066), sAPPα 5G11 (*Colombo et al., 2013*), sAPPβ 8C10 (*Kuhn et al., 2010*), APP 22C11 (Millipore, Billerica, MA), MAB348sAPPα (Covance, Princeton, NJ, Sig-39151), APP (Sigma-Aldrich, A8717), Cadherin 6 (Abcam/Epitomics, EPR217), Calnexin (Enzo, Stressgen, Farmingdale, NY, USA, ADI-SPA-860), Calbindin, (rabbit polyclonal anti-calbindin D-28k antibodies Swant, Bellinzona, Switzerland), mContactin (R&D Systems, Wiesbaden, Germany, AF4439), DNER (R&D Systems, AF2254), GFAP (Sigma-Aldrich, G3893), L1CAM Clone 555 (kindly provided by Peter Altevogt), LDL receptor (R&D systems, AF2255), Lrrc4b (R&D systems, AF4995), MT4-MMP (MMP17), (Abcam/Epitomics, EP1270Y), N-Cadherin (Abcam/Epitomics, EPR1791-4), NrCAM (Abcam, ab24344), Neuroligin-1 (Synaptic Systems, Göttingen, Germany 4C12), Synaptophysin (mouse monoclonal Sigma-Aldrich). Biotin-SP-conjugated goat anti-mouse secondary antibodies (Jackson Immunoresearch Laboratories, West Grove, Pa.), and Cy3-conjugated streptavidin (Dianova, Hamburg, Germany), Alexa 488 goat anti-rabbit secondary antibodies (Molecular Probes, Leiden, The Netherlands).

## ELISA

Levels of murine fractalkine (CX3CL1) were measured using an Elisa Kit (R&D systems, detection limit 98 pg/ml). Samples were stored at -80°C and analyzed undiluted after thawing.

## Mice

The generation and genotyping of knockout lines was formerly described: ADAM10 conditional KO embryos (*Gibb et al., 2010*), ADAM10 conditional KO mouse crossed with CamkII-Cre (*Prox et al., 2013*), APPsα-KI (*Ring et al., 2007*).

## Generation of Adam10 cKO, sAPPα knockin mice

Homozygous Adam10 deficient sAPPα knockin (sAPPαki) mice were generated from Adam10F/F CaMKIIα-Cretg/+ mice (*Prox et al., 2013*) crossed with APPsα-KI mice (*Ring et al., 2007*). Genotypes were confirmed by specific PCR on tail biopsies. PCR on CaMKIIα-Cre and floxed Adam10 was performed as previously described (*Prox et al., 2013*). PCR on sAPPα was performed with primers 5'GGCTGACAAACATCAAGACGGAAGAG3', 5'CACACCTCCCCCTGAACCTGAAAC3' and 5'CTGCGAGAGAGCATCCCTACAACC3'.

## Primary culture

Embryonic primary cortical neurons were prepared as described (*Kuhn et al., 2010*; *Colombo et al., 2013*). Briefly, cortex samples from ADAM10 conditional KO embryos (*Gibb et al., 2010*) were collected and dissociated in DMEM plus 200 U of papain (Sigma Aldrich). Neurons were plated in 6 well plates precoated with poly-D-lysine ($1.5\times10^6$ cells/well). Plating medium was 10% (v/v) fetal calf serum FCS/DMEM which was changed after 4 hr to B27/Neurobasal (Thermo, Braunschweig, Germany) supplemented with 0.5 mM glutamine and 1% P/S. All experimental procedures on animals were performed in accordance with the European Communities Council Directive (86/609/EEC).

## Virus production

Codon-improved Cre recombinase (iCre) (*Shimshek et al., 2002*) expressing lentiviral particles were generated as previously described (*Kuhn et al., 2010*; *Colombo et al., 2013*). Briefly, lentiviruses were generated by transient cotransfection of HEK293T cells (DSMZ, Braunschweig, Germany) with the plasmids psPAX2, pCDNA3.1(−)-VSV-G and as transfer vector F2UΔZeo-iCre using Lipofectamine 2000 (Thermo). Lentiviral particles for infection of murine primary cortical neurons were concentrated and purified by ultracentrifugation. Lentiviral stocks were stored at −80°C until use.

## SPECS

Secretome enriched and subsequent proteomic analyses were essentially performed as described earlier (*Kuhn et al., 2012*). In brief, 40 million neurons were plated. After two div, neurons were infected with concentrated lentiviruses coding for iCre or Control. After five div neurobasal medium was exchanged for neurobasal medium supplemented with 200 nM tetraacetyl-N-azidoacetyl mannosamine (ManNAZ). Two days conditioned media were subsequently collected and filtered through 0.45 µm PVDF Millex filter (Millipore, Darmstadt, Germany) into a VivaSpin 20 centrifugal concentrator (30 kDa) at 4°C. VivaSpin 20 columns were centrifuged at 4600 rpm at 4°C to remove non-metabolized ManNAZ. The retentate was filled with 20 ml H2O. This procedure was repeated three times. In the last step, the ddH2O refill step was omitted. Instead, 100 nM of DBCO-PEG12-biotin (Click Chemistry Tools, Scottsdale, AZ) diluted in 1 ml ddH2O was added to biotinylate metabolically azide-labelled glycoproteins. Centrifugal concentrators were incubated overnight at 4°C. For removal of non-reacted DBCO-PEG12-Biotin, VivaSpin20 centrifugal concentrators were subjected to three times of centrifugation with subsequent ddH2O refill. After the last centrifugation step, the retentate was diluted in 5 ml PBS with 2% SDS (v/v) and 2 mM Tris 2-carboxyethyl-phosphine (TCEP). For purification of biotinylated proteins, the sample was loaded on a 10-ml polyprep column with a streptavidin bead bed formed of 300 µl streptavidin slurry. After binding of proteins, streptavidin beads were washed three times with 10 ml PBS supplemented with 1% SDS. To elute the biotinylated and azide-labelled glycoproteins, streptavidin beads were boiled with urea sample buffer containing 3 mM biotin to compete for the binding of biotinylated proteins.

## SPECS-labeling of cellular glycoproteins

Neurons that were labeled with 200 nM ManNAz for 48 hr were washed twice with 5 ml cold PBS. Afterwards 100 nM DBCO-PEG12-biotin (Click Chemistry Tools) diluted in 2 ml PBS were evenly distributed on the neurons and incubated at 4°C for 2 hr. After the incubation period, neurons were washed twice with 5 ml PBS to be lysed in 5 ml STE buffer (NaCl, 150 mM, Tris 50 mM, 2 mM EDTA) with 1% (v/v) NP40 per flask. Lysates were subjected to a clarifying spin at 4000 g. The clarified lysate was subsequently loaded on a polyprep column with a streptavidin bead bed formed of 300 µl streptavidin slurry. After binding of proteins, streptavidin beads were washed three times with 10 ml PBS supplemented with 1% SDS. To elute the biotinylated and azide-labelled glycoproteins, streptavidin beads were boiled with urea sample buffer containing 3 mM biotin to compete for the binding of biotinylated proteins.

## Brain homogenates of conditional ADAM10 (AD10 fl/fl) CamkII-Cre mice

Brain homogenates were prepared as previously described (*Kuhn et al., 2012*) from P20 brains of conditional ADAM10 (AD10 fl/f) CamkII-Cre mice (*Prox et al., 2013*).

## Brain homogenates of ADAM10 cKO, sAPPα knockin mice

Mice were sacrificed at postnatal day P20 and brain regions were prepared (cortex, hippocampus and cerebellum). Samples were homogenized and lysed in cell lysis buffer (5 mM Tris base pH 7.4, 1 mM EGTA, 250 mM sucrose, 1% Triton X-100 supplied with protease inhibitors) by passing 15 times through a 23 G syringe and incubated for 1 hr at 4°C. Cell debris was pelleted and protein concentration in the supernatant was determined by BCA assay (Pierce, Thermo Fisher Scientific, Waltham, Ma). To analyze APP processing, samples were homogenized and lysed in RIPA buffer (50 mM HEPES pH 7.4, 150 mM sodium chloride, 1 mM EDTA, 0.1% SDS, 1% NP-40, 0.5 mM sodium deoxycholate supplied with protease inhibitors) by passing 15 times through a 23 G syringe. Cell debris was pelleted and supernatant was centrifuged (1 hr, 120000 x g, 4°C). Protein concentration in the supernatant was determined by BCA assay (Pierce). Samples were separated in either a 10% SDS-PAGE or a 4–12% gradient BIS/Tris NuPAGE Novex gels (Thermo Fisher Scientific) and transferred to nitrocellulose membranes (Roth, Karlsruhe, Germany) to perform Western blot analysis.

## Ultrastructural analysis of hippocampal sections

Ultrastructural sample preparation and analysis was performed as previously described (*Prox et al., 2013*).

## SDS–PAGE separation, trypsinization

Proteins were separated on a 10% Tris/glycine SDS gel. Afterwards, qualitatively equal gel slices were cut out from the gel with the exception of the remaining albumin band at around 60 kDa. Proteins in the gel slices were subject to trypsinization according to standard protocols (*Kuhn et al., 2015*).

## Western blot validation of substrates

For Western blot analysis of conditioned media and cell lysates of ADAM10 knockout neurons 20 µl of 48 hr conditioned supernatant (1.5 ml) of a six well plate with 1.5x106 neurons and 20 µl of 250 µl cell lysate of a six well plate with 1.5x106 neurons were loaded onto Tris/glycine SDS gels respectively. Proteins were transferred to 0,45 µm nitrocellulose membranes and blocked with skim milk prior to decoration with primary and secondary antibody. Six biological replicates were analyzed for each substrate.

## Mass spectrometric analysis

Mass spectrometry experiments were performed on an Easy NLC 1000 nanoflow HPLC system II (Proxeon, Odense, Denmark) connected to an LTQ-Velos Orbitrap Pro (Thermo Fisher Scientific, Braunschweig, Germany). Peptides were separated by reverse phase chromatography using in-house made 30 cm columns (New Objective, FS360-75-8-N-S-C30, Woburn, MA) packed with C18-AQ 2,4 µm resin (Dr Maisch GmbH, Ammerbuch-Entringen, Germany, Part No. r124.aq). A 90-min gradient (5–40%) at a flow rate of 200 nl/min was used. The measurement method consisted of an initial FTMS scan recorded in profile mode with 30000 m/z resolution, a mass range from 300 to 2000 m/z and a target value of 1000000. Subsequently, collision-induced dissociation (CID) fragmentation was performed for the 15 most intense ions with an isolation width of 2 Da in the ion trap. A target value of 10000, enabled charge state screening, a monoisotopic precursor selection, 35% normalized collision energy, an activation time of 10 ms, wide band activation and a dynamic exclusion list with 30 s exclusion time were applied.

## Analysis of mass spectrometry data

Considering that every biological replicate had been analyzed twice with mass spectrometry (technical replicate) after sample processing, five biological replicates with 2 technical replicates each of the neuronal secretome upon ADAM10 knockout and four biological replicates with two technical replicates each of the neuronal ADAM10 knockout surface proteome were analyzed with the freely available MaxQuant suite (version 1.4.1.2) (*Suzuki et al., 2012*). Protein identification was performed using the integrated Andromeda search algorithm (*Reiss et al., 2005*). First search, mass recalibration and main search of tryptic peptides were performed using a murine Uniprot database downloaded on the 08/21/2014 (86749 entries) for the ADAM10 secretome dataset and a murine Uniprot database downloaded on the 05/16/2014 (86749 entries) for the ADAM10 surfaceome dataset. Two missed cleavages were allowed. Peptide as well as protein false discovery rate was set to 1%. Mass accuracy was set to 20 ppm for the first search and 5 ppm for the main search. Quantification was performed between the respective control and ADAM10 knockout conditions on the basis of unique and razor peptides. Missing values were imputed in Perseus 1.5.16 following a standard distribution. p-values were calculated from log2 transformed relative intensity ratios of 5 biological replicates for the neuronal secretome and 4 biological replicates for the neuronal surface proteome with a heteroscedastic, two-sided student's t-test. Proteins with a p-value of p≤0.05 were considered as hits. To correct for multiple hypothesis testing the Benjamini-Hochberg post hoc was applied with an adjusted false-discovery rate of 0.10. MaxQuant output files (protein groups and peptides) are attached as *Supplementary file 3* for the neuronal secretome and the neuronal surface proteome. Calculated values on which the volcanos are based are attached as *Supplementary file 1* for the neuronal secretome and *Supplementary file 2* for the neuronal surface proteome. The mass spectrometry secretome raw data including MaxQuant output files have been deposited at the ProteomeXchange Consortium (http://proteomecentral.proteomexchange.org) via the PRIDE partner repository with the data set identifier PXD003426.

## Immunohistochemistry for synaptophysin

Animals were deeply anesthetized with chloral hydrate (7% in saline, intraperitoneally), and perfused intracardially with PBS followed by 4% paraformaldehyde in PBS. The brains were post-fixed overnight at 4°C in the same fixative. Sections 60 μm thick were cut using a vibratome and collected in PBS. The free floating sections were incubated 15 min in methanol/PBS 1:1 containing 1% $H_2O_2$ and blocked 45 min in a 5% BSA solution in PBS. Sections were then incubated with primary antibodies in PBS containing 0.25% BSA, 0.1% Triton X-100 and 0.05% $NaN_3$ for two nights at 4°C with gentle shaking. For the detection of synaptophysin, sections were incubated 90 min with biotinylated secondary antibody diluted in PBS, washed three times, and incubated 45 min with CyTM3-conjugated streptavidin. Then, sections were mounted on glass slides in PBS-glycerol 1:1 containing 0.1% Dapco (1,4-diazabicyclo [2.2.2.] octane, Sigma-Aldrich), and analyzed using an oil-immersion (HCX APO 63X/1.40 NA) objective coupled to a TCS SP5 confocal microscope (Leica, Bensheim, Germany). An experimenter not aware of the genotype of the mice was able to identify all mutants according to the synaptophysin immunostaining detected in the CA3 subfield of the hippocampus (*Montag-Sallaz et al., 2002*). For quantification of synaptophysin positve puncta, pictures of 164 x 164 μm (512x512 px, 320nm pixel width and height) from the CA3 region were taken. The stratum pyramidale was delineated with straight lines and analyzed with the counting particles option of ImageJ. The average area of small synaptophysin reactive puncta (0.64817 μm2 SD of 0.57921) representing inhibitory synapses was determined in wild-type mice and used to define the threshold for large puncta (average size of small puncta + 2x SD). Above this threshold ($\geq$1.80659 μm2) number and size of large immunoreactive puncta, representing mossy fiber terminals, were determined for wild-type and ADAM10-deficient mice.

## Lectin staining

Animals were perfused as described for the immunocytochemistry of synaptophysin. The brains were post-fixed 4 to 6 hr at 4°C in the same fixative, cryoprotected overnight in a solution of 30% sucrose in PBS, and frozen. Frontal olfactory bulb cryosections (60 μm thick) were collected in PBS. The free floating sections were incubated 15 min in methanol/PBS 1:1 containing 1% $H_2O_2$, blocked with a 2% BSA solution in PBS for 30 min, and then incubated for 2 hr with the plant lectin DBA conjugated to biotin (Dolichos Biflorus Agglutinin, Sigma) at 20 μg/ml in PBS and 0.25% Triton X-100. For the detection of the lectin, sections incubated 45 min with CyTM3-conjugated streptavidin. After washing in PBS, sections were mounted on glass slides with antifade media (Vectashield, Vector Labs, Burlingame, Ca) containing 4′,6-diamidino-2-phenylindole (DAPI; 1.5 μg/ml) and sealed. DAPI was used for nuclear staining. An experimenter not aware of the genotype of the mice was able to identify all mutants according to the lectin staining detecting aberrantly projecting axons in the olfactory bulb (*Montag-Sallaz et al., 2002*). On sections of the olfactory bulb from three animals per genotype, lectin-stained glomeruli (KO n = 110; WT n = 126) were classified as compact or diffuse. In WT, 89.2 ± 0.99% of all glomeruli were compact and 10.8 ± 0.99% diffuse, whereas in KO only 58.2 ± 9.1% were compact and 41.8 ± 9.4% diffuse.

## Statistics

P-values for Western blots and ELISA were calculated with log2 transformed relative values using a heteroscedastic, two-sided t-test. Immunohistochemical data were subjected to Analysis of Variance (ANOVA, factor genotype) and post hoc analysis (Scheffe's or Fisher PLSD) considering p<0.05 as significant.

## Acknowledgement

We thank Katrin Moschke, Daniela Hill and Dr. Rodrigo Herrera-Molina for excellent technical support. This work was supported by the DFG (SyNergy, FOR2290, SFB877-A3, Z3), the BMBF (RiMOD-FTD), the Verum Foundation, IWT, the Breuer Foundation Research Award, the Carl von Linde Junior Fellow Ship of the TUM Institute for Advanced Study. Deutsche Forschungsgemeinschaft Grants (MU 1457/9–1, 9–2 to UCM).

## Additional information

### Funding

| Funder | Author |
| --- | --- |
| Deutsche Forschungsgemeinschaft | Peer-Hendrik Kuhn<br>Alessio Vittorio Colombo<br>Ulrike Müller<br>Paul Saftig<br>Stefan F Lichtenthaler |
| Breuer Foundation | Stefan F Lichtenthaler |
| Stiftung VERUM | Stefan F Lichtenthaler |
| Technische Universitaet Muenchen, Institute for Advanced Study | Peer-Hendrik Kuhn |

The funders had no role in study design, data collection and interpretation, or the decision to submit the work for publication.

### Author contributions

P-HK, Conception and design, Acquisition of data, Analysis and interpretation of data, Drafting or revising the article; AVC, Acquisition of data, Analysis and interpretation of data, Drafting or revising the article; BS, SW, US, AL, EK, DM, UM, MS, PS, SB, Drafting or revising the article, Contributed unpublished essential data or reagents; DD, Acquisition of data, Analysis and interpretation of data, Drafting or revising the article, Contributed unpublished essential data or reagents; JH, Acquisition of data, Analysis and interpretation of data; SFL, Conception and design, Analysis and interpretation of data, Drafting or revising the article

### Author ORCIDs

Peer-Hendrik Kuhn, http://orcid.org/0000-0001-7891-5102
Dirk Montag, http://orcid.org/0000-0002-4964-1330

## Additional files

### Supplementary files

• Supplementary file 1. A10 knockout quantified secretome data set.

• Supplementary file 2. A10 knockout surfaceome dataset.

• Supplementary file 3. MaxQuant output files (protein groups and peptides) for the neuronal secretome and the neuronal surface proteome.

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
