## [Decision Letter]

Thank you for submitting your work entitled "Systematic substrate identification reveals a central role for the metalloprotease ADAM10 in axon and synapse function" for consideration by *eLife*. Your article has been reviewed by three peer reviewers, one of whom is a member of our Board of Reviewing Editors. This decision is taken by the reviewers after discussing their individual reviews with one another and the Reviewing Editor has drafted this decision to help you prepare a revised submission.

Overall, this is a very well written manuscript that presents data of the highest quality together with an excellent discussion of the results. This manuscript will make an important contribution to our understanding of the role of ADAM10, one of the major sheddases, in the brain. The unbiased approach taken by Kuhn et al. is extremely valuable, since it not only highlights the breadth of the substrate repertoire of ADAM10 in the brain (which is not unexpected), but also identifies some of the top candidates, and addresses the very important concept of the relative contribution to different sheddases to the processing of individual substrates. Anyone studying the role of proteolysis in the context of an intact organism requires this information to take a more holistic and general approach towards understanding the role of proteases and of substrate processing in development and disease. This study sets the stage for further analysis of the role of ADAM10 various processes that are regulated by specific substrates in the brain, and provides a clear outline of how this can be accomplished in other cell types. As such, this timely study should be of significant interest to the readers of *eLife*.

We raise below a series of minor and more important criticisms, but we do not request additional experimentation to address these. We are satisfied that the issues raised can be addressed in a written form in the rebuttal letter and partially in the revised version of the final manuscript.

1) Regarding the title, perhaps the word "reveals" should be replaced with "indicates" or "suggests" or "underscores"?

2) In the last sentence of the Abstract, please consider including the words indicated by arrowheads. "In summary, the novel ADAM10 substrates provide a >potential< molecular basis for neuronal network dysfunctions in conditional ADAM10-/- mice and demonstrate a fundamental function of ADAM10 >as a sheddase< in the brain."

3) Subheading “Analysis of mass spectrometry data”: please explain what a technical replicate is.

4) Paragraph four, Discussion; paper of Zhou et al. (2012) should be cited as well as (Colombo et al., 2012).

5) The authors might consider to include one manuscript (Barao et al. Cell Rep. 2015 Sep 1;12(9):1367-76.) as it nicely complements some of the statements in their Discussion. For instance when they claim "whether CHL1 and L1 cleavage by ADAM10 or BACE1 has different functional outcomes, has not been investigated" is not right. Barao et al. investigated whether ADAM10 (metalloprotease) inhibition affects growth cone collapse via CHL1 processing: had no effect, while BACE1 inhibition blocked growth cone collapse (Barao et al., Cell reports, 2015). This paper also confirms that CHL1 is cleaved by BACE1 and at least one metalloprotease in neurons. The paper deserves here and further below some discussion as it complements nicely the statements in this paper.

6) It might help the readers to mention that for instance Notch, a major substrate of ADAM10, was not identified in the proteomic screen. This is likely explained by the fact that this processing is ligand dependent, a situation not mimicked in the cell culture conditions.

7) Table 1: previous substrates are not indicated in bold.

8) The grey scale on the pie graphs (Figure 1, Figure 3) is difficult to follow, please use colors or more distinct grey shading, or place the label next to the pieces of the pie chart.

9) Can you please briefly explain how "false discovery rate multiple hypothesis testing" is done, in one or two sentences?

10) In Figure 2, the efficient deletion of ADAM10 should be shown first.

11) The results in Figure 5 are very interesting and could be presented in more detail. It would also be good to include some quantification, if possible. Specifically, regarding the images of the glomeruli in the olfactory epithelium, it appears that the structure of the glomeruli is less defined and condensed in the mutants than control. Is this generally the case? Moreover, regarding the mis-targeted projections, would it be possible to quantify this observation, for example by counting the number of mis-targeted projections per glomerulus on a number of slides of mutant and control animals? Likewise, the results depicted in the image of the hippocampus are actually quite compelling, but not immediately apparent at first glance, at least to this reviewer. Figure 5 perhaps merits its own figure, with two separate parts for the olfactory glomeruli and the hippocampus, higher magnification images of the hippocampus, and some arrows pointing towards the increased staining in the pyramidal somata in the mutant mice. This qualitative finding does not necessarily require quantification, but should be illustrated more clearly.

12) The authors performed a series of experiments to clarify, which of the newly identified substrates could be cleaved by ADAM10 as well as by the β-secretase BACE. This is an important point. However, the authors should also considered that the changes they observe could be also, in part, explained by the upregulation in the Adam10 cKO neurons of other ADAM family members, such as for example ADAM17. It should be easy to verify (or exclude) if this is the case.

13) Neuronal cultures were established from E15-16 cortex. At this stage cortical neurogenesis is still ongoing and largely dependent on Notch signaling. Since the Notch receptor is one of the well-characterized Adam10 substrate, have the authors checked the composition of iCRE treated and untreated neuronal cultures during the culture period? This is an important point because a differential proportion of proliferating vs. differentiated cell types, or among differentiated neurons might influence the cultures' secretome. This could be easily checked using proliferation (BrdU incorporation p-PH3) and neuronal differentiation (Cux1, Pax6, Tbr1 etc) markers.

14) Among other findings, the authors show that the membrane-tethered metalloproteinase MT4-MMP is a highly processed Adam10 substrate. Although they state in the Discussion that MT4-MMP mostly processes extracellular matrix molecules, can they exclude that some of the changes could be an indirect effect of MT4-MMP processing? This point could be experimentally addressed using specific inhibitors, if available. Alternative this point should be addressed in the Discussion.

15) The authors have focused their attention on type I membrane glycoproteins, which are indeed significantly changed as shown in Figure 1. However, there is an even more evident change among secreted proteins. Indeed, they found only 13 secreted proteins in the secretome of Adam10 cKO neurons vs the 113 found in the controls (Figure 1). This finding deserves attention and should be addressed. Do the authors know the nature of the secreted proteins? Does Adam10 act on secreted proteins? Is this a reflection of the culture composition as mentioned in point 2?

16) The biochemical analysis in Figure 2 and Figure 4 is impeccable. Data shown in Figure 5 are less so. Blots show multiple bands that are unclear and molecular weights are variable. It will be nice if they could improve these data or at least expand the figure legend for the less expert in the field.

17) The analysis of axon targeting and synapse organization in the hippocampus and olfactory system is quite skimpy. Since the authors use this in vivo analysis to support their proteomic analysis, why do not concentrate just on one or the other region but offering a better characterization?

18) The absence/strong reduction of Adam10 expression (Figure 2) in the presence of iCRE could be perhaps moved to Figure 1 next to the experimental design.

19) The strength of this manuscript is its accurate proteomic analysis but the Discussion does not exploit this point enough. The authors should consider refocusing the Discussion and highlight better their findings.

20) The authors have identified SLIRTK among the targets of ADAM10 function. The authors may consider discussing the possible implication of this molecule (perhaps in relation to Robo receptors) in olfactory axon mistargeting.

21) First paragraph of the Introduction: the sentence "….to shape the cell surface by controlling the length (?), and thus the function…." is unclear. The length of what?

---

## [Author Response]

*1) Regarding the title, perhaps the word "reveals" should be replaced with "indicates" or "suggests" or "underscores"?*

The title word ‘reveals’ has been replaced with ‘indicates’. The new title reads: “Systematic substrate identification indicates a central role for the metalloprotease ADAM10 in axon targeting and synapse function.”

*2) In the last sentence of the Abstract, please consider including the words indicated by arrowheads. "In summary, the novel ADAM10 substrates provide a >potential< molecular basis for neuronal network dysfunctions in conditional ADAM10-/- mice and demonstrate a fundamental function of ADAM10 >as a sheddase< in the brain."*

We included “potential” and “as a sheddase” in the last sentence of the Abstract. The sentence now reads as suggested by the reviewers.

*3) Subheading “Analysis of mass spectrometry data”: please explain what a technical replicate is.*

To explain this, we changed the sentence to:

“Considering that every biological replicate had been analyzed twice with mass spectrometry (technical replicate) after sample processing, five biological replicates with 2 technical replicates each of the neuronal secretome upon ADAM10 knockout and four biological replicates with two technical replicates each of the neuronal ADAM10 knockout surface proteome were analyzed with the freely available MaxQuant suite (version 1.4.1.2).”

*4) Paragraph four, Discussion; paper of Zhou et al. (2012) should be cited as well as (Colombo et al., 2012).*

We added the citation of Zhou et al., 2012.

*5) The authors might consider to include one manuscript (Barao et al. Cell Rep. 2015 Sep 1;12(9):1367-76.) as it nicely complements some of the statements in their Discussion. For instance when they claim "whether CHL1 and L1 cleavage by ADAM10 or BACE1 has different functional outcomes, has not been investigated" is not right. Barao et al. investigated whether ADAM10 (metalloprotease) inhibition affects growth cone collapse via CHL1 processing: had no effect, while BACE1 inhibition blocked growth cone collapse (Barao et al., Cell reports, 2015). This paper also confirms that CHL1 is cleaved by BACE1 and at least one metalloprotease in neurons. The paper deserves here and further below some discussion as it complements nicely the statements in this paper.*

We included a passage that carefully puts CHL1 cleavage by BACE1 and Sema3a mediated axon repulsion in context with our findings:

“While for L1 differential functional outcomes of ADAM10 and BACE1 cleavage have not been investigated so far, it recently has been shown that CHL1 cleavage by BACE1 is indispensible for Semaphorin 3a induced axon repulsion while inhibition of metalloprotease mediated CHL1 cleavage had no impact which suggests that modulation of axon growth in the axonal or presynaptic compartment via CHL1 is only modulated by BACE1.”

*6) It might help the readers to mention that for instance Notch, a major substrate of ADAM10, was not identified in the proteomic screen. This is likely explained by the fact that this processing is ligand dependent, a situation not mimicked in the cell culture conditions.*

We agree with the reviewers that there may be different reasons why Notch1 was not detected. Thus, we included the following new sentence:

“For example Notch1, an established ADAM10 substrate, was not detected in our analysis, potentially because its expression level was below the detection limit or because it was not expressed in the neurons at the time-point of analysis or because its ADAM10-mediated shedding is ligand-dependent.”

7) Table 1: previous substrates are not indicated in bold.

Protein names of previously identified substrates are now highlighted in bold, italic. Gene Symbols of substrates validated by immunoblot are highlighted in bold.

*8) The grey scale on the pie graphs (*Figure 1, Figure 3) is difficult to follow, please use colors or more distinct grey shading, or place the label next to the pieces of the pie chart.

We now used colors with more contrast to improve visibility. This should also help if the PDF is printed in gray colors.

*9) Can you please briefly explain how "false discovery rate multiple hypothesis testing" is done, in one or two sentences?*

False discovery rate was calculated according to the method of Benjamini and Hochberg: We changed the sentence to:

“When applying false-discovery rate-based multiple hypothesis testing according to the method of Benjamini and Hochberg considering an FDR=0.1 and all identified glycoproteins as hypotheses, 46 membrane proteins remain significantly reduced (Figure 1 and Table 1)”.

This information is also given in the Methods section (subheading “Cross-talk between ADAM10 and BACE1”).

*10) In Figure 2, the efficient deletion of ADAM10 should be shown first.*

ADAM10 is now shown first in Figure 2.

*11) The results in Figure 5 are very interesting and could be presented in more detail. It would also be good to include some quantification, if possible. Specifically, regarding the images of the glomeruli in the olfactory epithelium, it appears that the structure of the glomeruli is less defined and condensed in the mutants than control. Is this generally the case? Moreover, regarding the mis-targeted projections, would it be possible to quantify this observation, for example by counting the number of mis-targeted projections per glomerulus on a number of slides of mutant and control animals? Likewise, the results depicted in the image of the hippocampus are actually quite compelling, but not immediately apparent at first glance, at least to this reviewer. Figure 5 perhaps merits its own figure, with two separate parts for the olfactory glomeruli and the hippocampus, higher magnification images of the hippocampus, and some arrows pointing towards the increased staining in the pyramidal somata in the mutant mice. This qualitative finding does not necessarily require quantification, but should be illustrated more clearly.*

Indeed, many glomeruli of the mutant appear less compact. We have quantified this observation and included it in the new figure (Figure 5). The number of mis-targeted axons stained with the lectin DBA is low, however, in wild-type they are not observed at all in adult animals. We have improved the figure showing large synaptophysin-positive puncta surrounding pyramidal cell bodies in the stratum pyramidale of the mutant indicating aberrant mossy fiber terminals and quantified these (Figure 5).

*12) The authors performed a series of experiments to clarify, which of the newly identified substrates could be cleaved by ADAM10 as well as by the β-secretase BACE. This is an important point. However, the authors should also considered that the changes they observe could be also, in part, explained by the upregulation in the Adam10 cKO neurons of other ADAM family members, such as for example ADAM17. It should be easy to verify (or exclude) if this is the case.*

We agree that this is a concern and we are aware of cross-talk between different proteases. Thus, we have carried out the suggested experiment, which is already published in one of our earlier papers (Kuhn et al. EMBO J 2010). We knocked-down ADAM10, which nearly completely abolished APP α-secretase cleavage. Using qRT-PCR we found that the expression of ADAM17 and ADAM9 was not increased in the absence of ADAM10. The neuronal cultures in the previous study and in our current study were identical. Thus, we feel that a compensatory upregulation of ADAM17 – which is the closest homolog to ADAM10 and another major sheddase – is not a relevant mechanism upon ADAM10 loss-of-function in our neuronal cultures. Hence, we included the sentence: Another important finding was that in contrast to other proteases like ADAM17 or ADAM9 (Kuhn et al., ADAM10 is the physiologically relevant, constitutive α-secretase of the amyloid precursor protein in primary neurons. The EMBO journal. 2010;29(17):3020-32), cellular levels of the GPI-anchored, neuronally expressed metalloprotease MT4MMP increased while MT4MMP ectodomain shedding was abolished upon Adam10 deletion (Rikimaru et al. Establishment of an MT4-MMP-deficient mouse strain representing an efficient tracking system for MT4-MMP/MMP-17 expression in vivo using β-galactosidase. Genes Cells. 2007;12(9):1091-100) (subheading “Immunohistochemistry for synaptophysin”).

*13) Neuronal cultures were established from E15-16 cortex. At this stage cortical neurogenesis is still ongoing and largely dependent on Notch signaling. Since the Notch receptor is one of the well-characterized Adam10 substrate, have the authors checked the composition of iCRE treated and untreated neuronal cultures during the culture period? This is an important point because a differential proportion of proliferating vs. differentiated cell types, or among differentiated neurons might influence the cultures' secretome. This could be easily checked using proliferation (BrdU incorporation p-PH3) and neuronal differentiation (Cux1, Pax6, Tbr1 etc) markers.*

Thank you very much for this thoughtful comment. In Figure 1 we included the marker β III tubulin which is only expressed in mature primary neurons. We did not observe any significant alteration of β III tubulin expression after ADAM10 knockout when quantifying β III tubulin expression in 4 biological replicates. Furthermore, we did not observe major changes in the neuronal surface proteome which additionally speaks in favor that neuronal differentiation is not affected in the in vitro context. In the Results section we included the following sentence:

“Staining for β III tubulin suggests that knockout of ADAM10 did not alter neuronal differentiation of primary neurons in vitro (Figure 1).”

*14) Among other findings, the authors show that the membrane-tethered metalloproteinase MT4-MMP is a highly processed Adam10 substrate. Although they state in the Discussion that MT4-MMP mostly processes extracellular matrix molecules, can they exclude that some of the changes could be an indirect effect of MT4-MMP processing? This point could be experimentally addressed using specific inhibitors, if available. Alternative this point should be addressed in the Discussion.*

At this point little is known about MT4-MMP substrates, but it appears to mostly cleave soluble proteins of the extracellular matrix (Itoh Matrix Biology 2015). However, as we cannot exclude that reduced MT4-MMP shedding (which results in a consequential increase in surface bound MT4-MMP and reduced soluble MT4-MMP) may potentially contribute to an altered secretome in ADAM10-/- neurons, we included the sentence:

“MT4MMP is assumed to mostly cleave soluble proteins in the extracellular matrix (4). However, since little is known about the physiological substrate spectrum of MT4MMP in neurons, we cannot exclude that some of our observed changes in the ADAM10-/- neurons and their secretome may be an indirect consequence of the reduced MT4MMP shedding.”

Unfortunately no specific MT4-MMP inhibitors are available. To slightly shorten the Discussion (as suggested in point 20 below), we deleted several lines in this paragraph describing a potential role of MT4MMP in aneurysms.

*15) The authors have focused their attention on type I membrane glycoproteins, which are indeed significantly changed as shown in Figure 1. However, there is an even more evident change among secreted proteins. Indeed, they found only 13 secreted proteins in the secretome of Adam10 cKO neurons vs the 113 found in the controls (Figure 1). This finding deserves attention and should be addressed. Do the authors know the nature of the secreted proteins? Does Adam10 act on secreted proteins? Is this a reflection of the culture composition as mentioned in point 2?*

We are sorry to not have clarified this point well enough in the initial version of the manuscript. All 113 identified secreted proteins in the controls have also been identified in the ADAM10-/- secretome. Among these 113 identified secreted proteins only 13 were significantly altered upon ADAM10 deletion. Given that ADAM10 is a membrane-bound sheddase all identified and verified substrates are membrane proteins. Thus, it is expectable that the secretion of only few soluble proteins would be changed in ADAM10-/- cells. And these proteins are not expected to be direct substrates of ADAM10, but to be changed as an indirect consequence of ADAM10-deletion. The number of soluble proteins identified in wild type and ADAM10 knockout mice was the same. However, few of these soluble proteins were quantitatively reduced in ADAM10 knockout mice most likely due to stabilization of their receptors at the neuronal membrane or due to their reduced expression.

*16) The biochemical analysis in Figure 2 and Figure 4 is impeccable. Data shown in Figure 5 are less so. Blots show multiple bands that are unclear and molecular weights are variable. It will be nice if they could improve these data or at least expand the figure legend for the less expert in the field.*

We have expanded the figure legend for new Figure 6 to improve the readability of the data.

*17) The analysis of axon targeting and synapse organization in the hippocampus and olfactory system is quite skimpy. Since the authors use this in vivo analysis to support their proteomic analysis, why do not concentrate just on one or the other region but offering a better characterization?*

We have revised the figure showing the axonal mis-targeting phenotype and included a quantification (new Figure 5). The CA3 hippocampal region and the olfactory glomeruli were chosen because they represent clearly structured brain areas where a mis-targeting can be demonstrated. The same procedure was applied previously to characterize axonal mis-targeting in several other mutants e. g. NCAM, CHL1, NrCAM, and BACE1.

*18) The absence/strong reduction of Adam10 expression (Figure 2) in the presence of iCRE could be perhaps moved to Figure 1 next to the experimental design.*

We feel that ADAM10 deletion should be shown directly next to the substrate blots, in agreement with reviewer comment 10 above. Thus, we kept it in Figure 2.

*19) The strength of this manuscript is its accurate proteomic analysis but the Discussion does not exploit this point enough. The authors should consider refocusing the Discussion and highlight better their findings.*

We increased the discussion of our proteomic data. The new paragraph now reads as:

“Here, we provide the first systematic analysis of ADAM10 substrates in primary cortical neurons of a conditional Adam10 knockout mouse. […] The large number of substrates makes ADAM10 a major sheddase in the nervous system and an important modifier of the neuronal secretome.”

Accordingly, we shortened the discussion of the substrate MT4MMP and of Notch.

*20) The authors have identified SLIRTK among the targets of ADAM10 function. The authors may consider discussing the possible implication of this molecule (perhaps in relation to Robo receptors) in olfactory axon mistargeting.*

This is a very good suggestion. However, we decided to not include such a discussion for two reasons. First, according to point 19 (above) we have increased the Discussion about the proteomics parts of the study and thus needed to shorten other parts accordingly. Second, while the role of Slit proteins and their interaction with Robo is indeed well established for axon targeting, the role of SLITRKs is much less understood and a clear role in axon targeting has not yet been demonstrated.

*21) First paragraph of the Introduction: the sentence "*…*.to shape the cell surface by controlling the length (?), and thus the function*…*" is unclear. The length of what?*

The sentence has been changed into: This process, referred to as ectodomain shedding, is a mechanism that shapes the cell surface by controlling the ectodomain length of cell surface membrane proteins thus modulating their function.